# Emergence of orbital angular moment at van Hove singularity in graphene/*h*-BN moiré superlattice

Rai Moriya [1✉], Kei Kinoshita[1], J. A. Crosse[2], Kenji Watanabe [3], Takashi Taniguchi [1,4], Satoru Masubuchi [1], Pilkyung Moon [2,5], Mikito Koshino[6] & Tomoki Machida [1✉]

Bloch electrons lacking inversion symmetry exhibit orbital magnetic moments owing to the rotation around their center of mass; this moment induces a valley splitting in a magnetic field. For the graphene/*h*-BN moiré superlattice, inversion symmetry is broken by the *h*-BN. The superlattice potential generates a series of Dirac points (DPs) and van Hove singularities (vHSs) within an experimentally accessible low energy state, providing a platform to study orbital moments with respect to band structure. In this work, theoretical calculations and magnetothermoelectric measurements are combined to reveal the emergence of an orbital magnetic moment at vHSs in graphene/*h*-BN moiré superlattices. The thermoelectric signal for the vHS at the low energy side of the hole-side secondary DP exhibited significant magnetic field-induced valley splitting with an effective g-factor of approximately 130; splitting for other vHSs was negligible. This was attributed to the emergence of an orbital magnetic moment at the second vHS at the hole-side.

---

[1] Institute of Industrial Science, University of Tokyo, 4-6-1 Komaba, Meguro, Tokyo 153-8505, Japan. [2] New York University Shanghai and NYU-ECNU Institute of Physics at NYU Shanghai, Shanghai, China. [3] Research Center for Functional Materials, National Institute for Materials Science, 1-1 Namiki, Tsukuba 305-0044, Japan. [4] International Center for Materials Nanoarchitectonics, National Institute for Materials Science, 1-1 Namiki, Tsukuba 305-0044, Japan. [5] State Key Laboratory of Precision Spectroscopy, East China Normal University, Shanghai 200062, China. [6] Department of Physics, Osaka University, Toyonaka, Osaka 560-0043, Japan. ✉email: moriyar@iis.u-tokyo.ac.jp; tmachida@iis.u-tokyo.ac.jp

**B**erry curvature and orbital magnetic moment are two pseudovectors that are observed in the presence of spatial inversion asymmetry. These enable us to control valley contrasting phenomena in two-dimensional materials[1–3]; therefore, they are essential in the field of valleytronics. The presence of the Berry curvature induces electrons to have anomalous velocity perpendicular to an applied electric field, thus exhibiting the valley Hall effect. The valley Hall effect has been demonstrated at the gapped Dirac band such as gapped bilayer graphene and monolayer graphene/*h*-BN moiré superlattice[4–6]. In contrast to this, an orbital magnetic moment will induce an energy shift under the application of the perpendicular magnetic field; this is known as valley Zeeman splitting and has been demonstrated in gapped bilayer graphene[7–9] as well as transition metal dichalcogenides[10]. Both Berry curvature and orbital magnetic moment, in principle, strongly depends on the local structure of the band. Here we choose graphene/hexagonal-boron nitride (*h*-BN) as a model system to investigate the orbital magnetic moment in relation to the band structure.

## Results

**Band structure and orbital magnetic moment in graphene/*h*-BN.** A pristine monolayer graphene is an inversion symmetric crystal (Fig. 1a). At the corner of the hexagonal Brillouin zone, different bands are connected at the Dirac point (DP) located at the K- and K′-point, and the energies between different valleys (K and K′) are degenerated. When graphene is transferred on a *h*-BN insulator through van der Waals force, two significant changes occur in the band structure. First, the interaction between graphene and *h*-BN breaks the spatial inversion symmetry of graphene. Second, the lattice mismatch between two crystals generates a moiré pattern (Fig. 1b). Owing to the moiré superlattice potential, the band structure of graphene is completely reconstructed[11–13], as shown in Fig. 1c, d, and calculated at the K-point using an effective continuum theory[12]. Herein, to emphasize the effect of inversion asymmetry, two different calculations are performed: one that holds spatial inversion symmetry (inversion symmetric model shown in Fig. 1c) and one that uses an inversion asymmetric (Fig. 1d) model (see Supplementary Note 1). The inversion asymmetric model was demonstrated to comprehensively match the experimental results on band structure as well as Landau quantization[11–16]. The plot of energy with respect to the density of states (DOS) derived from Fig. 1d is shown in Fig. 1e. Because of the moiré potential, series of sub-DPs, such as secondary-DPs (SDPs) or tertiary-DPs (TDPs), are generated at both low- and high-energy sides of the main DP[11–13,17–19]. These result in local minima in the DOS (Fig. 1e). Moreover, saddle points between the DPs, known as van Hove singularities (vHSs), are represented as local maxima. The most striking effect of inversion asymmetry indicated in Fig. 1c, d is the opening of the band gap at several points[11–13], which is indicated by the red- and blue-dashed circles. First, a band gap is opened at the main DP (i.e., the K-point in the first band) and both electron- and hole-side SDPs (i.e., the X-point; blue-dashed circles in Fig. 1d). This gapped DP has been reported to exhibit a finite Berry curvature[4–6].

In this study, we observed that there was another point (i.e., a second vHS in the hole-side located at the Y-point (red-dashed circle in Fig. 1d)) that exhibited a pronounced effect of inversion asymmetry; this point has not been discussed to date. In the symmetric model (Fig. 1c), this point is a point of contact between the hole-side of the second and third bands. Inversion asymmetry induces a significant gap opening; as a result, this point becomes the vHS point in the inversion asymmetric model. This is a unique property of this particular vHS because the influence of inversion symmetry on other vHSs from the first to the third bands is limited. The opening of a gap with inversion asymmetry naturally results in the generation of an orbital magnetic moment **m**(**k**) at this vHS. This is an analog of a spin angular moment of the electron. However, **m**(**k**) is orbital in

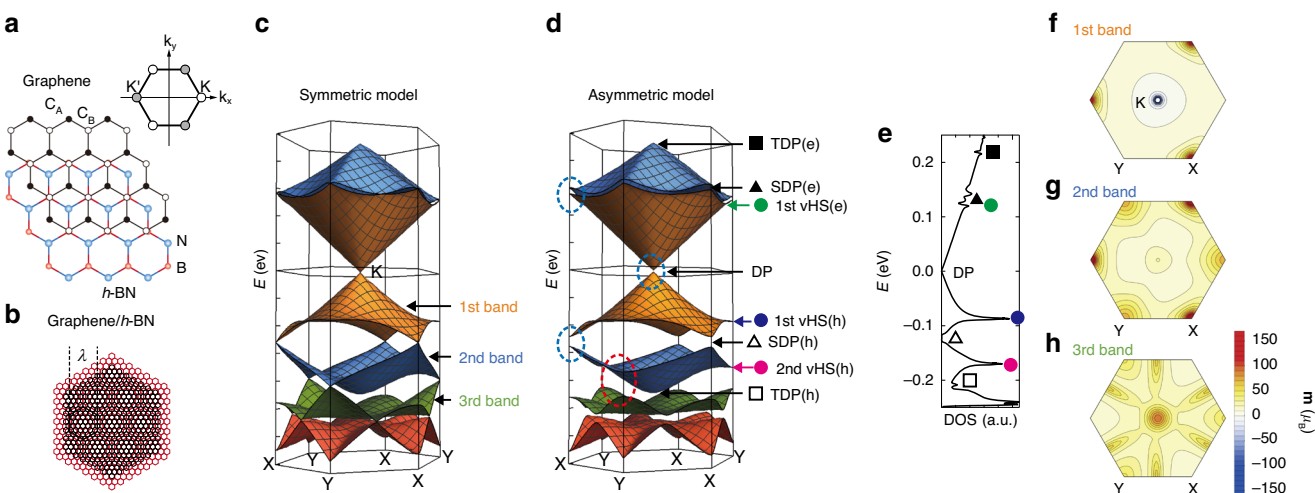

**Fig. 1 Orbital magnetic moment in graphene/*h*-BN moiré superlattice. a** Crystal structure of graphene and *h*-BN. The inset depicts the corresponding k-space of the Fermi surface of graphene. **b** Moiré pattern generated in the aligned graphene/*h*-BN heterostructure. The period of moiré potential λ is indicated. Here, the lattice constant of *h*-BN is drawn 20% larger than that of graphene to enhance the visibility of the moiré pattern. **c, d** Band structure of the graphene/*h*-BN moiré superlattice with θ = 0° at the K-point calculated using an effective continuum model. Calculation is performed under an **c** inversion symmetric model and an **d** inversion asymmetric model. The band structure exhibits series of bands—the first, second, and third bands—located at the hole-side of the DP. In addition to the main DP, series of secondary (SDPs) and tertiary DPs (TDPs) are observed in the electron- (e) and hole- (h) sides of the main DP; these are indicated by solid black square (electron-side TDP), solid black triangle (electron-side SDP), open black triangle (hole-side SDP), and open black square (hole-side TDP), respectively. In addition, series of vHS points are observed in between the DPs; these are indicated by solid green circle (electron-side first vHS), solid blue circle (hole-side first vHS), and solid magenta circle (hole-side second vHS), respectively. **e** Relationship between energy E and density of states (DOS) of aligned graphene/*h*-BN derived from band structure in **d**. **f–h** Calculated orbital angular moment **m**(**k**) for hole-side **f** first band, **g** second band, and **h** third band.

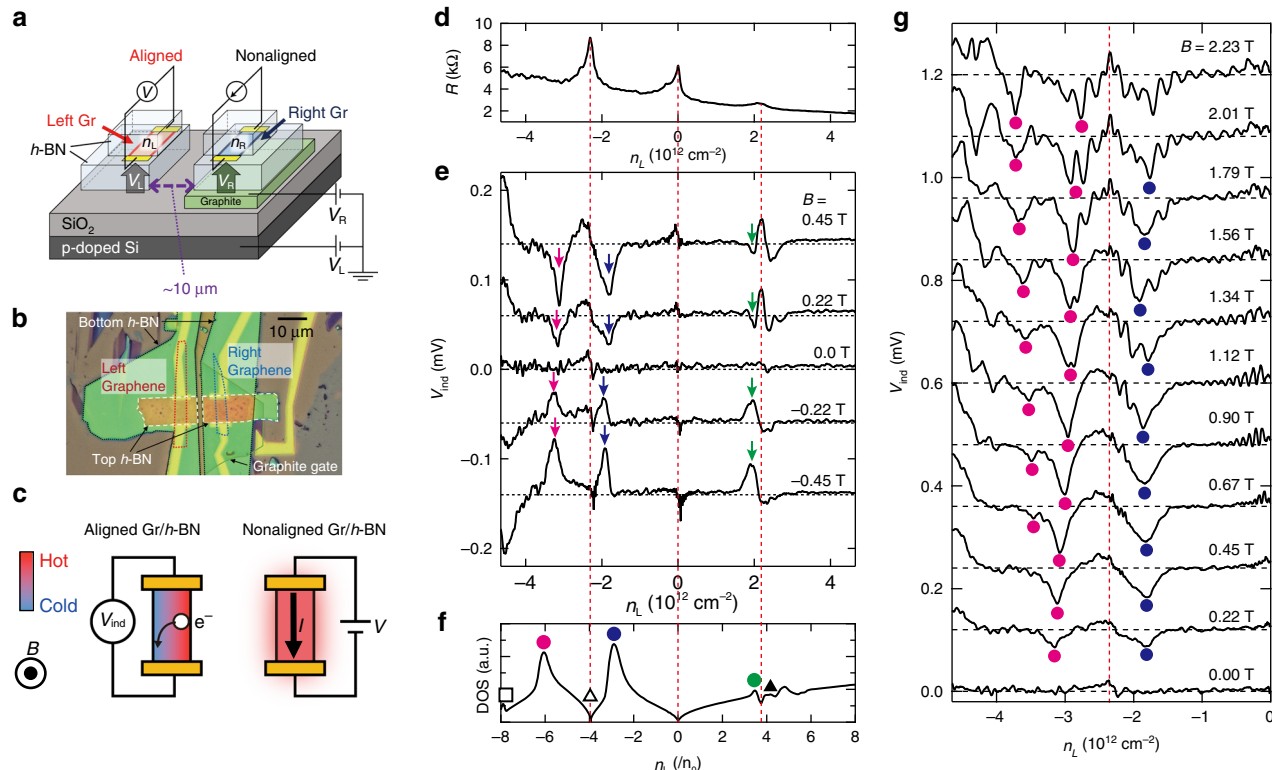

**Fig. 2 Magnetothermoelectric measurement of graphene/h-BN moiré superlattice. a** Schematic of the device structure. The lattice of the graphene (Gr) on the left is aligned with the underlying h-BN to achieve a moiré pattern ($\theta \sim 0°$). As the graphene on the right is not aligned with the h-BN lattice, the influence of moiré is negligible. The graphene on the right has a graphite gate (GrG) underneath, which can apply a local back-gate voltage of $V_R$ to control its carrier density $n_R$; the carrier density of the graphene on the left $n_L$ is controlled using the back-gate voltage $V_L$. **b** Optical micrograph of fabricated device. **c** Schematic illustration of thermoelectric voltage detection in a parallel graphene device. **d** Two-terminal resistance $R$ of the graphene on the left as a function of its carrier density $n_L$ at 2.0 K. **e** Thermoelectric voltage $V_{ind}$ generated in the graphene on the left as a function of its carrier density $n_L$, measured at a different out-of-plane magnetic field $B = \pm0.45$, 0.22, and 0 T. A constant power $P$ of 1 mW is applied to the graphene on the right. Traces are offset for clarity and offset is depicted by black dashed lines. Signals appear at a position away from the SDPs are indicated by the magenta, blue, and green arrows. **f** Calculated DOS with respect to normalized carrier density $n/n_0$ derived from Fig. 1e. Definition of solid and open circles, triangles, and square are same as Fig. 1e. **g** $V_{ind}$ as a function of the carrier density $n_L$ measured at a different out-of-plane magnetic field $B$ obtained from the graphene/h-BN moiré superlattice at 2.0 K. Here, the positions of signal dips are indicated by solid magenta circles (second hole-side vHS) and solid blue circles (first hole-side vHS), respectively. The dashed vertical red line depicts the carrier density of the SDP(h). Traces are offset for clarity and offset is depicted by black horizontal dashed lines.

nature, and it is responsible for the splitting between the K and K′ valley instead of the up- and down-spin. A general form of the **m**(**k**) is expressed as the following equation[2,3,9]:

$$\mathbf{m}(\mathbf{k}) = -\mathrm{i}(e/2\hbar) \times \langle \nabla_\mathbf{k} u | \times [H(\mathbf{k}) - \varepsilon(\mathbf{k})] | \nabla_\mathbf{k} u \rangle, \quad (1)$$

where $|u(\mathbf{k})\rangle$ is the periodic part of the Bloch function, $H(\mathbf{k})$ is the Bloch Hamiltonian, and $\varepsilon(\mathbf{k})$ is the dispersion of band. We calculated **m**(**k**) for the band structure in Fig. 1d and obtained **m**(**k**) with respect to the wavevector of the first, second, and third bands in the hole-side of the main DP, as depicted in Fig. 1f–h, respectively. **m**(**k**) is large for the gapped DPs, including both the main DP and SDPs[3]. Moreover, the calculation shows that there is a noticeably large **m**(**k**) presented at the second vHS(h; i.e., the Y-point in the second band). The second vHS(h) yields an **m**(**k**) value of 66 μB, corresponding to a valley g-factor of ~130. This value is noticeably large as an orbital magnetic moment of vHS, considering that **m**(**k**) is negligibly small at other vHSs such as Y-point in the first band (see Supplementary Note 2 for the detail comparison).

**All-electrical magnetothermoelectric measurement**. To verify the existence of orbital moment at the vHS, we performed

magnetothermoelectric measurement, as schematically illustrated in Fig. 2a; the optical micrograph of the device is shown in Fig. 2b. Two different flakes of graphene encapsulated with h-BN were placed on the SiO2/doped-Si substrate. We created trench in the h-BN to reduce capacitive coupling between the two graphene flakes (see Methods section and Supplementary Note 3). The graphene depicted on the right has a graphite gate (GrG), which can apply a local back-gate voltage of $V_R$ to control the carrier density $n_R$. The carrier density of the graphene on the left $n_L$ is controlled using the back-gate voltage $V_L$. The lattice of the graphene on the left is aligned with the underlying h-BN substrate, to ensure a large period of moiré potential. In contrast, the period of the graphene on the right is small because of the significant rotational misalignment between graphene and h-BN. As illustrated in Fig. 2c, the nonaligned graphene is used as a local heater by applying current to induce Joule heating. The heat radiated from the nonaligned graphene induces heating of the aligned graphene, thereby creating a temperature gradient perpendicular to the channel of the aligned graphene. In the presence of this temperature gradient and the magnetic field perpendicular to the plane $B$, thermoelectric voltage $V_{ind}$ is generated along the channel direction of the aligned graphene, due to the Nernst effect[20–24]. This voltage can be detected using the metal electrode

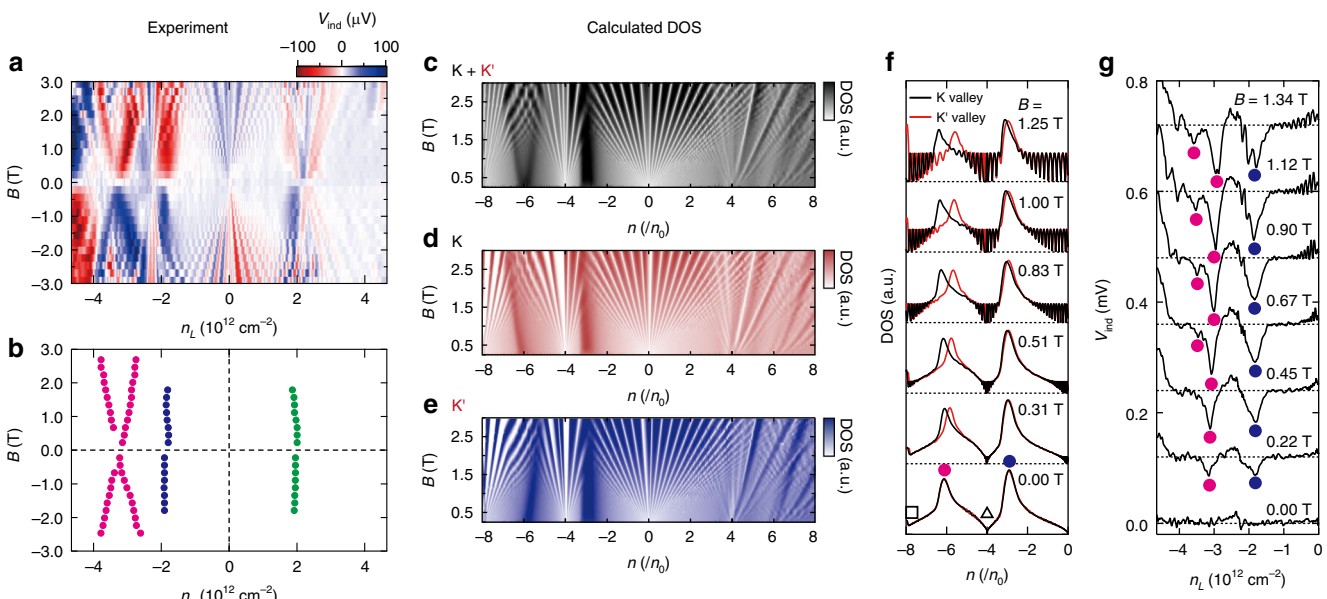

**Fig. 3 Valley splitting of the thermoelectric voltage at van Hove singularity. a** $V_{ind}$ as a function of $n_L$ and $B$. **b** Peak positions of $V_{ind}$ at each vHS obtained from **a** plotted as a solid magenta circle (second hole-side vHS), solid blue circle (first hole-side vHS), and solid green circle (first electron-side vHS). **c–e** Calculated DOS as a function of the normalized carrier density $n/n_0$ and $B$ under the condition of **c** total DOS including both K and K' valley, **d** DOS of K valley, and **e** DOS of K' valley. **f** DOS with respect to the normalized carrier density $n/n_0$ calculated at different magnetic field values. Traces are offset for clarity and offset is depicted by black dashed lines. The positions of DPs and vHSs are indicated by solid magenta circle (second hole-side vHS), solid blue circle (first hole-side vHS), open square (holes-side TDP), and open triangle (hole-side SDP), respectively. **g** $V_{ind}$ as a function of the carrier density $n_L$ measured at a different out-of-plane magnetic field $B$ measured at 2.0 K. Traces are offset for clarity and offset is depicted by black dashed lines. These are same data as Fig. 2g.

connected at both ends of the graphene. The thermopower is significantly enhanced at the vHS as it is demonstrated by photo-Nernst measurement[25]. Therefore, our all-electrical magneto-thermoelectric measurement provides sensitive probes for vHSs.

The two-terminal resistance $R$ of the aligned graphene is measured under a sweep of $n_L$ at 2.0 K; the results are plotted in Fig. 2d. In addition to the resistance peak caused by the main DP at $n_L = 0$, other peaks from the SDP at the electron- and the hole-side at $n_L = +2.20$ and $-2.25 \times 10^{12}$ cm$^{-2}$, respectively, are visible. Although its origin is not clear at this moment, there is a slope background in $R$ that increases towards the left in the figure; we infer this is related to the inhomogeneity of metal/graphene contact. Based on these $n_L$ values, the period of moiré potential $\lambda$ is determined using the relationship $n_{SDP} = 8/(\sqrt{3}\lambda^2)$ where $n_{SDP}$ represents the carrier density at the resistance peak position of the SDP[14–16]. Using the obtained $\lambda$ value of ~14 nm, we determined the lattice alignment angle between graphene and $h$-BN as $\theta \sim 0°$[17]. During thermoelectric measurement, a constant power $P = 1$ mW is applied to the nonaligned graphene using a source meter, while maintaining its carrier density at charge neutrality $n_R = 0$ at $B = 0$ and quantum Hall filling factor $\nu = -6$ such that $n_R = 6eB/h$ in the magnetic field, where e depicts electron charge and $h$ the Planck constant. This maintain the two-terminal resistance of heater graphene to be kept ~10 kΩ throughout the measurement from no magnetic field to high magnetic field (see Supplementary Note 4). Under this condition, the increase in the average temperature of the aligned graphene is estimated to be ~10 K. $V_{ind}$ as a function of $n_L$ is measured at different magnetic fields $B$, and the results are presented in Fig. 2e by the solid black lines. For $B = 0$ T, the $V_{ind}$ signal is negligibly small. The small signal around $n_L = +2.20$ and $-2.25 \times 10^{12}$ cm$^{-2}$ at $B = 0$ T can be attributed to a Seebeck effect of SDP, originating from an asymmetry of the contacts. On increasing $B$, initially, the

$V_{ind}$ signals that appeared at the main DP and the SDP had similar amplitudes. Thereafter, the $V_{ind}$ signal at the DPs exhibits a peak in positive $B$ and a dip in negative $B$. For comparison, the $V_{ind}$ signal obtained from nonaligned graphene/$h$-BN device is provided in Supplementary Note 5. In addition, a larger signal appears at a position away from the SDPs, as indicated by the arrows in Fig. 2e. Evidently, the signs of the values of $V_{ind}$ at these locations differ from those of the DPs, such that the signal from the three DPs exhibit dips in negative $B$ values, whereas the signals indicated by the arrows exhibit peaks. These results are compared to the calculated DOS derived from Fig. 1e plotted versus carrier density $n$, as depicted in Fig. 2f. Here, the carrier density was calculated by integrating DOS such that $n(E_F) = \int_0^{E_F} DOS(E)dE$ The unit of this axis is normalized using the total number of electron states divided by the area of a filled Bloch band $n_0$. Therefore, $n/n_0 = \pm 4$ and 8 correspond to the SDP and TDP, respectively. A comparison of Fig. 2d–f indicates that additional large signals of $V_{ind}$ are observed at three vHSs, namely the first vHS(e), first vHS(h), and second vHS(h). The first vHSs on both the electron- and hole-side had previously been investigated[25]; here, we investigated the second vHS(h) which was observed[26,27] but not discussed before. In addition to these, it is noteworthy that the $V_{ind}$ signal largely increases around $n_L = -4.65 \times 10^{12}$ cm$^{-2}$ in Fig. 2e due to the presence of TDP, the location of which can be seen from Fig. 2f; however, a detail investigation of this TDP is difficult on this device since the demonstration of carrier density higher than that of Fig. 2e is limited by the breakdown of SiO$_2$ dielectric.

The $V_{ind}$ versus $n_L$ are measured in a magnetic field from $B = +0$ to 2.23 T and results are shown in Fig. 2g. The magnetothermopower from the second vHS(h) and first vHS(h) exhibits a considerably different behavior. Their positions are depicted by a solid magenta circle and a blue circle, respectively, in Fig. 2g. The dip-shape signal generated from the second vHS

(h) exhibits significant splitting with increasing $B$ (see also refs. [26,27]), whereas the $V_{ind}$ signals from other vHSs, i.e., the first vHS(h) and the first vHS(e) (see Fig. 3a, b), did not exhibit any evident splitting with increasing $B$.

**Valley splitting of the thermoelectric voltage at vHS.** To investigate the origin of the magnetic field-induced splitting in $V_{ind}$ at the second vHS(h), we present a comparison between the experimental values and the calculated DOS in aligned graphene/$h$-BN obtained from the continuum model[12]. In Fig. 3a, we depict the plot of $V_{ind}$ as a function of $n_L$ and $B$ plotted for $+B$ and $-B$. The signals arising from three vHSs—the first vHS(e), first vHS (h), and second vHS(h)—are extracted from the data, and plotted with green, blue, and magenta solid circles, respectively in Fig. 3b (details of the extraction procedure as well as the $V_{ind}$ data in high magnetic field region are presented in Supplementary Note 6). We present a calculated DOS as a function of $n/n_0$ and $B$ in Fig. 3c–e. The calculated DOS as a function of energy $E$ is also provided in Supplementary Note 7. Here, we plotted the total DOS, including the DOS for both K and K′ valleys (Fig. 3c), DOS for K valley (Fig. 3d), and DOS for K′ valley (Fig. 3e), separately. A good coincidence was observed between Fig. 3a, c; the $B$-induced splitting of vHSs as well as the Landau fan diagram of the DPs were completely reproduced by the calculation. By plotting the K- and K′-valley contributions to the DOS separately in Fig. 3d, e, we observed that the DOS peak at the second vHS(h) for the K(K′) valley changed to the larger (smaller) $n/n_0$ value with the magnetic field. Furthermore, we plotted individual traces of Fig. 3d, e in Fig. 3f, for a comparison with the experimental data of $V_{ind}$ shown in Fig. 3g (The data plotted in Fig. 3g is same data as Fig. 2g). These data also exhibited good agreement with each other. Therefore, we conclude that the magnetic field induced splitting of the second vHS(h) indicated by our results is caused by the large magnetic field-induced valley splitting.

## Discussion

The significant valley splitting corresponding to a valley g-factor of ~130 extracted from Fig. 1g indicates the emergence of a large $\mathbf{m}(\mathbf{k})$ at the second vHS(h). To obtain a finite $\mathbf{m}(\mathbf{k})$ from Eq. 1, the matrix element for the inter sub-band coupling should be non-zero[28] (see Supplementary Note 8 for a detailed discussion). This can only be achieved at a point of contact between the bands (i.e., DP) in the graphene/$h$-BN under the inversion symmetric model. As seen from the comparison between Fig. 1c, d, the second vHS(h) in Fig. 1d is a point of contact between hole-side second and third band at Y-point under inversion symmetry; thus, it is a sub-DP. This sub-DP is gapped under the introduction of inversion asymmetry and becomes a vHS point. Therefore, the inter sub-band coupling at this vHS point is still non-zero, even after the opening of the gap. This hidden DP nature of the Y-point at the hole-side second and third band demonstrates the emergence of an orbital magnetic moment at the second vHS(h) in the graphene/$h$-BN moiré superlattice. Our all-electrical magnetothermoelectric measurements unveil this unique band property of graphene/$h$-BN moiré superlattice. Since there is a growing interest in twisted bilayer or twisted multilayer graphene[29–36], this method may also provide a sensitive probe to detect orbital magnetic moment with respect to different band structures of such new material systems.

## Methods

**Sample fabrication.** A device was fabricated using sequential dry release transfer of an individual flake. Two different flakes of monolayer graphene were successively transferred on top of the $h$-BN/graphite structure on the SiO$_2$/doped-Si substrate. Finally, both flakes of graphene were encapsulated with another $h$-BN. The flake transfer was performed using a poly(propylene) carbonate (PPC)-based dry release transfer method[37,38], and the thickness of $h$-BN was chosen to be

~30–40 nm. Using e-beam (EB) lithography and reactive ion etching employing a mixture of CF$_4$ and Ar gases, a trench was fabricated in the $h$-BN located between the two graphene flakes. Thus, the connection through $h$-BN between the two graphene flakes was eliminated. Finally, a contact pattern was created using an EB lithography with PMMA resist, and a 90-nm-thick Au/5-nm-thick Cr stack was deposited using EB evaporation. It should be noted that during device fabrication, the device was annealed at 350 °C under an Ar/3% H$_2$ atmosphere after each flake transfer. This annealing caused the transferred graphene flakes to rotate within the plane toward the preferred orientation with respect to the adjacent $h$-BN[39,40]. In the device presented in Fig. 2a, b, the graphene on the left is almost perfectly aligned with $h$-BN, to generate moiré superlattice potential; however, the graphene on the right was not aligned.

**Experimental setup.** A liquid He-cooled variable temperature cryostat equipped with a superconducting magnet was used for transport measurement. The differential resistance of graphene was measured by applying an AC current $I_{ac} = 10$ nA, with a frequency of 18 Hz; the AC voltage was measured using a lock-in amplifier. A source measure unit (Keithley 2400) was used to apply constant power to the heater graphene.

## Data availability

The datasets generated during and/or analyzed during the current study are available from the corresponding author on reasonable request.

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

## Acknowledgements

This work was supported by CREST, Japan Science and Technology Agency (JST) (grant number JPMJCR15F3), and JSPS KAKENHI (grant numbers JP19H02542, JP19H01820, JP20H00127, JP20H00354, and JP20H01840). J.A.C. and P.M. acknowledge the support of NYU Shanghai (Start-Up Funds), NYU-ECNU Institute of Physics at NYU Shanghai, and New York University Global Seed Grants for Collaborative Research. J.A.C. acknowledges the support from the National Science Foundation of China (NSFC; Grant No. 11750110420). P.M. acknowledges the support from the Science and Technology Commission of Shanghai Municipality (STCSM; Grant No. 19ZR1436400).

## Author contributions

R.M. and T.M. conceived the experiment. K.K. fabricated the device. M.K. developed the theoretical aspects. K.K. and R.M. performed the measurements with the help of S.M. J.A.C. and P.M. performed the theoretical calculation and analyzed the results. T.T. and K.W. grew the h-BN crystals. R.M. and T.M. wrote the paper with input from all authors. All authors contributed to extensive discussions of the results.

## Competing interests

The authors declare no competing interests.
