## [Peer Review File · Nature Communications]

REVIEWER COMMENTS

Reviewer #1 (Remarks to the Author):

This manuscript reports a very complex and interesting experiment aimed at detecting topological magnetic moments related to the Berry phase/curvature specific to moire superlattice minibands edges for electrons in aligned graphene/hBN heterostructures. The results are publishable in NC, as the topic is of a general interest and these observations have potential to attract interest in a very broad topological materials community. I recommend to accept this manuscript, upon a minor revision taking into account the following points.

1. The topological properties of states at the miniband edges in moire superlattices in graphene/hBN heterostructures have been pointed out before theoretically in Phys. Rev. B 87, 245408 (2013) and Annalen der Physik, 527, 259 (2015) [here, specifically for the minibands in Hofstadter spectrum], with the experimental evidence for Dirac-like spectrum obtained by a peculiar sequences of miniband Landau levels given published in Nature 487, 594 (2013). This should be boldly acknowledged and cited in the introduction of this manuscript and compared with its conclusions (rather than mentioning in passing in the supplementary material).

2. The topological magnetic moment, with the valley g-factor values $g \sim 40-140$, has been studied in gapped bilayer graphene theoretically [Phys. Rev. B 98, 155435 (2018)] and observed experimentally in point contact spectroscopy of gapped bilayers [Phys. Rev. Lett. 121, 257702 (2018); Phys. Rev. Lett. 124, 126802 (2020)]. This should be acknowledged and cited in the introduction of this manuscript, correcting a vague statement 'Bloch electrons ... experimental proof of this intrinsic property of bands is difficult to obtain' in the first paragraph and an inaccurate statement on page 1 [I quote: 'A presence of the Berry curvature has been demonstrated ... experimental proof of the orbital magnetic moment is difficult to obtain.'].]

3. The presented data lack information on the temperature dependence of the reported thermoelectric effect. If such data are available, I recommend to include them in the supplementary section.

Reviewer #2 (Remarks to the Author):

In their article "Emergence of orbital angular moment at van Hove singularity in graphene/h-BN Moiré superlattice", Moriya and co-workers describe a magneto-thermoelectric measurement of graphene that is aligned with hBN. A nearby graphene sheet is heated by 1mW. At finite magnetic field, a voltage is induced in the moiré structure due to the Nernst effect. A large signal is observed at the main and the secondary dirac peaks and at the Van-Hove singularities. As a function of perpendicular magnetic field, the signal at the 2nd VHS splits, which the authors attribute to a magnetic field-induced valley splitting with a g-factor of ~ 140 .

The authors manage to characterize band properties of the hBN/Graphene moiré superlattice by applying an unconventional method with interesting technological aspects. As the magnetothermoelectric measurements are also relevant to the optical community (photo-Nernst effect), I believe that the manuscript can appeal to a rather broad readership. Given the growing interest in graphene superlattices, especially twisted bilayer graphene, the hereby presented method to access band properties is certainly timely.

There are two main aspects of novelty.

- To my knowledge, this is the first time that an (all-electrical) magneto-thermoelectric measurement is applied to a moiré superlattice structure. However, the authors should state explicitly that the photo-Nernst effect has been exploited to reveal VHS and 2nd DPs in graphene/hBN in an optical setup. Ref.20 is in that respect a closely related work and the similarities/differences have to be described.

- Second, the authors are able to measure the valley g-factor at the 2nd VHS. Whereas it is unclear if this has any technological relevance, it still demonstrates the strength of the method used. But this aspect has to be worked out more clearly by comparing to the literature. Are there any signatures of the appearance and splitting of the 2nd VHS in high-quality magneto-transport experiments? What other methods exist to measure the valley-g-factor and how do they compare to the one presented here? The most closely related work is Ref.14. Also here a more explicit comparison is needed (e.g. I think that the method of Ref.14 is more accurate as it can be employed at base temperature, but it requires low densities (not possible at the 2nd VHS) and more complex geometries).

A concluding section with the outlook on twisted bi- and multilayer graphene structure would be interesting to the readers. What kind of features in the complex band structures of these materials could be potentially detected with this setup?

Overall, the manuscript is carefully written and well-structured. The science is accurately described and presented in an understandable way.

I have a few questions regarding the scientific aspects:

- The authors induce a temperature gradient by heating a close-by graphene sheet. Is it possible to provide quantitative simulations? How does heat transfer work through SiO₂? How important is it that the hBN is cut in the middle?

- If a power of 1mW is applied, how stable is the temperature? Is there some equilibration time needed before measuring?

- At B=0, there is a wiggle at $n_L = -4n_0$ and a small one at $+4n_0$. What is the origin of this signal?

- There is the statement on page 5 that the valley-g-factor is "unusually" large. How does this $g \sim 140$ compare to other works? Has a valley-g-factor on Moiré/hBN ever been measured before?

- Please state explicitly how you distinguish the signal at the 2nd VHS from Landau-levels (Fig.3a)

- Page 7: "The signals arising from three vHSs—the first vHS(e), first vHS(h), and second vHS(h)—are extracted from the data, as shown in Fig. 3(b)". How was this extraction done? State explicitly.

- How strongly were the parameters in the continuum model adjusted in order to fit the experimental data?

- Why is the semi-classical model that is used to extract the orbital magnetic moment still valid at large magnetic field, i.e. in the presence of Landau levels?

Minor comments

- Check font sizes in Figures

- If traces are offset from each other for clarity, this should be state explicitly

- Figure 1fgh: Are these the same colors as in d? It is difficult to link the plots to the points in the band structure and the label is tiny.

- Do Fig. 2g and 3g show the same data? If yes, that should be stated explicitly.

- Page 7: "The variations in V_{ind} with respect to n_L measured using ...". What is the precise meaning of the "variation in V_{ind} "?

Reviewer #3 (Remarks to the Author):

The authors report a large orbital magnetic moment at the second hole van Hove singularity in graphene/hBN moire superlattice. The main evidence is the magnetic field splitting of Nernst effect peaks at vHs, which agrees with model calculation. It provides interesting insight into the band structure of graphene/hBN moire structure. A few comments/questions below:

1. Can the authors include more detail about the device while generating heat, e.g. what is the carrier density and channel resistivity? What is the contact resistance? This is relevant because it affects where the hot spot is.

2. The authors mentioned that "The opening of a large gap with inversion asymmetry naturally results in the generation of an orbital magnetic moment $m(k)$ at this (second) vHS". Is the main Dirac point also gapped within the inversion asymmetry model? If yes, is it understood why a similar effect reported for the second vHs is not observed for the first hole and electron side vHs? Would it be possible to provide an intuitive picture of what leads to the exceptionally large orbital magnetic moment in the second hole vHs?

3. In figure 1d, what is causing the slope background in the resistance? In figure 1e, V_{ind} seems to diverge on the left end of the plot, what is the reason for that? In figure 3g, the V_{ind} signal for the right red dot is always stronger than the left dot, does this indicate one valley has larger DOS than the other? Can the authors comment on this?

4. Can the authors show in a separate plot either in the SI or main text, another version of Figure 3 c-e with horizontal axis being energy, with which the actual $m(k)$ value can be more easily extracted? I also recommend including the details of how to convert energy to density in the calculation.

5. Can the authors explain the inclusion of the inversion-symmetric model in the manuscript? Previous experiments, as cited in this manuscript, already point to the inversion-asymmetric models.

6. Typo. The letter "m" in "moire" should not be in caps.

Thanks for reviewing our manuscript titled “Emergence of orbital angular moment at van Hove singularity in graphene/h-BN moiré superlattice”. We are very glad that the reviewers gave us very positive comments. As an attached file, please find our replies to each of the comments. We would like to note that all changes in the revised manuscript file was indicated with red colored text.

Reviewer: 1

General comment)

This manuscript reports a very complex and interesting experiment aimed at detecting topological magnetic moments related to the Berry phase/curvature specific to moire superlattice minibands edges for electrons in aligned graphene/hBN heterostructures. The results are publishable in NC, as the topic is of a general interest and these observations have potential to attract interest in a very broad topological materials community. I recommend to accept this manuscript, upon a minor revision taking into account the following points.

Reply for general comment)

We appreciate the reviewer for his/her positive comment for our manuscript. Please find our replies to each of your comments.

Comment 1)

The topological properties of states at the miniband edges in moire superlattices in graphene/hBN heterostructures have been pointed out before theoretically in Phys. Rev. B 87, 245408 (2013) and Annalen der Physik, 527, 259 (2015) [here, specifically for the minibands in Hofstadter spectrum], with the experimental evidence for Dirac-like spectrum obtained by a peculiar sequences of miniband Landau levels given published in Nature 487, 594 (2013). This should be boldly acknowledged and cited in the introduction of this manuscript and compared with its conclusions (rather than mentioning in passing in the supplementary material).

Reply for comment 1)

Thanks for suggesting these papers. We agree that suggested two papers [1,2] were pioneering works that demonstrated miniband structure and Hofstadter spectrum in moiré superlattice. In the revised manuscript, these references are included in the main text during discussion of comparing theoretical calculation with experimental results.

Briefly, the suggested two papers [1,2] are more similar to the symmetric model in our manuscript since these papers treated that inversion asymmetry is small therefore no gap opening at main- and sub-DPs. The paper written by Moon *et al.* [3] considering the case that inversion asymmetry is not negligibly small in graphene/h-BN moiré superlattice; this is corresponding to the case for inversion asymmetric model in our manuscript, and due to the inversion asymmetry there is a gap opening in main- and sub-DPs. Except for this difference, other features such as miniband structure as well as topological properties contain lots of common understanding between ref. [1,2] and ref. [3]; thus we believe that both papers are equally important to discuss in the manuscript.

Comment 2)

The topological magnetic moment, with the valley g-factor values $g \sim 40-140$, has been studied in gapped bilayer graphene theoretically [Phys. Rev. B 98, 155435 (2018)] and observed experimentally in point contact spectroscopy of gapped bilayers [Phys. Rev. Lett. 121, 257702 (2018); Phys. Rev. Lett. 124, 126802 (2020)]. This should be acknowledged and cited in the introduction of this manuscript, correcting a vague statement

'Bloch electrons ... experimental proof of this intrinsic property of bands is difficult to obtain' in the first paragraph and an inaccurate statement on page 1 [I quote: 'A presence of the Berry curvature has been demonstrated ... experimental proof of the orbital magnetic moment is difficult to obtain.'].
Reply for comment 2)

The reviewer is right that the presence of orbital magnetic moment around the charge neutrality of biased bilayer graphene was proposed and experimentally demonstrated in the manuscripts suggested by the reviewer. We agree that these previous works need to be cited correctly and acknowledged in the main text. The orbital magnetic moment had been intensively studied theoretically in gapped Dirac points such as biased bilayer graphene and transition metal dichalcogenide monolayer. Experimental verification of the concept is also presented in these systems; thus, discussion on these previous works are added in the revised manuscript. We included these new reference papers in the revised manuscript together with more clear comparisons with these results.

We also revised following sentences since because of above-mentioned background, these sentences are not appropriate as reviewer also suggested.

'Bloch electrons ... experimental proof of this intrinsic property of bands is difficult to obtain'

→ Bloch electrons lacking inversion symmetry exhibit orbital magnetic moments owing to the rotation around their center of mass; the presence of this intrinsic property of bands exhibits valley Zeeman splitting.

'A presence of the Berry curvature has been demonstrated ... experimental proof of the orbital magnetic moment is difficult to obtain.'

→ A presence of the Berry curvature induces anomalous velocity perpendicular to an applied electric field, thus exhibits valley Hall effect. The valley Hall effect has been demonstrated at the gapped Dirac band such as gapped bilayer graphene and monolayer graphene/h-BN moiré superlattice [4-6]. In contrast to this, an orbital magnetic moment will induce energy shift in the magnetic field; that is valley Zeeman splitting [7-9]. Both Berry curvature and orbital magnetic moment, in principle, strongly depends on the local structure of the band.

Comment 3)

The presented data lack information on the temperature dependence of the reported thermoelectric effect. If such data are available, I recommend to include them in the supplementary section.

Reply for comment 3)

It is interesting to see how temperature influences our observation. We found that our measurement scheme of using a graphene heater is particularly efficient to induce heat gradient in another graphene in low temperature (such as ~2.0 K in our experiment) and becomes inefficient in higher temperatures. Therefore, in the higher temperature, the thermoelectric signal decreases mainly due to the reduction of heat transfer from heater graphene. To capture the right physics, we need to maintain the same temperature gradient in the moiré graphene; thus we need more heater power at a higher temperature. This was rather challenging at this moment, simply the power injected into the heater graphene is already reasonably large (~1 mW) and an increase of injected power will eventually damage the heater graphene. As we explained in the answer in (Reply for comment 8 of Reviewer 2), the current device structure was optimized to obtain good signals at low temperatures. We need to revise the structure for accurate determination of the temperature dependence of the phenomena. We are still working on these points and willing to present such results in future work.

In addition to the above-mentioned changes, there are other two corrections in the manuscript that we would like to ask you to check. These corrections have appeared when we consider the question from the Reviewer 3.

[Correction 1]

(Statement in the submitted manuscript) We stated that during magneto-thermoelectric effect measurement, heater graphene's carrier density is tuned to its charge neutrality (DP) under the application of current for Joule heating.

(Corrected statement) During magneto-thermoelectric effect measurement, heater graphene's carrier density n_R is tuned to the quantum Hall filling factor $\nu = -6$ such that $n_R = 6eB/h$ during the application of current for Joule heating. This means that carrier density at zero magnetic field is charge neutrality $n_R = 0$ same as previous manuscript, however $n_R = 6eB/h$ under the application of magnetic field.

The intension of both methods is to keep heater graphene's condition to be constant. Keeping graphene's carrier density to the quantum Hall filling factor $\nu = -6$ keeps graphene heater's total resistance to be around 10 k Ω (channel resistance + contact resistance). The detail of this can be found in (Reply to comment 1) section for Reviewer 3. We found this mistake when we considered questions from reviewer 3. We deeply apologize for this; this correction was also mentioned to the other two reviewers. We are very glad that if you could check this revised manuscript.

[Correction 2]

We also wanted to correct valley g-factor at the 2nd hole-side vHS from $g \sim 140$ (previous manuscript) to $g \sim 130$ (in the revised manuscript). This is because since during the initial submission of the manuscript, we determined the amplitude of orbital magnetic moment $m(k)$ from the color scale of the contour plot as shown in Figure R1(b,c,d) below. Then we determined $m(k) \sim 70 \mu_B$ and this corresponding to valley g-factor of $g = 2m(k) = 140$. During the consideration of comment from reviewer 3, we noticed that a more precise value of calculated $m(k)$ is $m(k) = 66.4 \mu_B$ as you can see from Figure R1(e,f,g). In Figure R1(e,f,g), we plotted calculated $m(k)$ values along Y-point to X-point in k-space. The $m(k) = 66.4 \mu_B$ could be approximated as $m(k) \sim 70 \mu_B$, however, it is not appropriate to claim that $g \sim 140$ based on this. Since the more precise g-factor value is $g = 2m(k) = 132.8$, we think it is more correct to state that determined valley g-factor is $g \sim 130$. Because of these reasons, we would like to correct $g \sim 140$ to $g \sim 130$ in the revised manuscript. We believe that this correction does not have a significant influence on the main claim of our manuscript; however, we would like to let all three reviewers know about this change. We are very glad if you could consider this correction in addition to all the replies to your comment.

Figure R1: (a) Band structure of the graphene/h-BN Moiré superlattice with $\theta = 0^\circ$ at K-point calculated using an effective continuum model. Calculation is performed under an inversion asymmetric model. (b,c,d) Calculated orbital angular momentum m for hole-side (b) first band, (c) second band, and (d) third band. (e,f,g) Line profile of m vs. k between Y-point to X-point for (e) first band, (f) second band, and (g) third band.

Reviewer: 2

General comment)

In their article “Emergence of orbital angular moment at van Hove singularity in graphene/h-BN Moiré superlattice”, Moriya and co-workers describe a magneto-thermoelectric measurement of graphene that is aligned with hBN. A nearby graphene sheet is heated by 1mW. At finite magnetic field, a voltage is induced in the moiré structure due to the Nernst effect. A large signal is observed at the main and the secondary dirac peaks and at the Van-Hove singularities. As a function of perpendicular magnetic field, the signal at the 2nd VHS splits, which the authors attribute to a magnetic field-induced valley splitting with a g-factor of ~140.

The authors manage to characterize band properties of the hBN/Graphene moiré superlattice by applying an unconventional method with interesting technological aspects. As the magnetothermoelectric measurements are also relevant to the optical community (photo-Nernst effect), I believe that the manuscript can appeal to a rather broad readership. Given the growing interest in graphene superlattices, especially twisted bilayer graphene, the hereby presented method to access band properties is certainly timely.

Reply for general comment)

Thanks for your comment. We agree with the reviewer that our method could be also useful in the community of twisted bilayer graphene and other related structures for investigating their band properties. Please find our replies to each of your comments.

Comment 1)

There are two main aspects of novelty.

- To my knowledge, this is the first time that an (all-electrical) magneto-thermoelectric measurement is applied to a moiré superlattice structure. However, the authors should state explicitly that the photo-Nernst effect has been exploited to reveal VHS and 2nd DPs in graphene/hBN in an optical setup. Ref.20 is in that respect a closely related work and the similarities/differences have to be described.

Reply for comment 1)

Thanks for your comment. The ref. 20 “Multiple hot-carrier collection in photo-excited graphene Moiré superlattices.” demonstrated the photo-Nernst effect for secondary DPs and some of the vHSs for the first time. Here we demonstrated the detection of Nernst effect by all-electrical setup. We investigated vHSs in high carrier density range that is outside of the range investigated in ref. 20. We certainly agree that the finding of ref. 20 is quite influential to the achievement of our study. The mechanism of generating a signal in both our experiment and ref. 20 is the thermoelectric effect; this is similar. The difference in the experimental setup is how to introduce temperature gradients within the device; that could cause a difference in device structure and measurement scheme.

Technically, we believe that the advantage of using the photo-Nernst effect (ref. 20) can be the simplicity of the device structure since the heated area of the graphene can be determined by the size of the laser spot used for photoexcitation. Such a measurement could become more difficult for very low temperatures and very high magnetic fields where one needs a sophisticated setup to manage optical measurement under such an environment. The all-electrical measurement may be easier in such extreme conditions (low T and high B) since this, in principle, only requires a system designed for electrical measurement. However, the device structure can be more complicated so that device fabrication is more difficult.

In our opinion, the photo-Nernst effect can be also used for detecting the valley splitting of vHSs. In the revised version, we emphasize that point such that thermoelectric measurement is critical for our observation of orbital moments and either electrical or optical measurement can be used for detecting this.

Comment 2)

- Second, the authors are able to measure the valley g-factor at the 2nd VHS. Whereas it is unclear if this has any technological relevance, it still demonstrates the strength of the method used. But this aspect has to be worked out more clearly by comparing to the literature. Are there any signatures of the appearance and splitting of the 2nd VHS in high-quality magneto-transport experiments? What other methods exist to measure the valley-g-factor and how do they compare to the one presented here? The most closely related work is Ref.14. Also here a more explicit comparison is needed (e.g. I think that the method of Ref.14 is more accurate as it can be employed at base temperature, but it requires low densities (not possible at the 2nd VHS) and more complex geometries).

Reply for comment 2)

Thanks for your comment. We believe that the reviewer's concern is the detailed comparisons of our measurement scheme and literature.

Regarding the first question “Are there any signatures of the appearance and splitting of the 2nd VHS in high-quality magneto-transport experiments?”.

I checked through literature that was showing measurement data sets of secondary Dirac points and vHSs in graphene/h-BN or bilayer graphene/h-BN moiré superlattice systems. We think some of them seem to show signals like our observation of the splitting in the 2nd hole-side vHS. The data from these papers (ref. [4,5]) are presented in Figure R2. Data from other papers do not clearly show such a signal [6-8], even though all of these papers are performing similar type four-terminal resistance or Hall resistance measurement. None of the above-mentioned papers contains discussions and comments about the signal around 2nd hole-side vHS we described in our paper. Further, no discussion about the possible magnetic field splitting of vHSs.

Our opinion is that the signal from the vHS should be always present, in principle. The vHS gives apparent sign changes in Hall resistance since it is the point for a change of majority carrier between electron

(a) R. Krishna Kumar, X. Chen, G. H. Auton, A. Mishchenko, D. A. Bandurin, S. V. Morozov, Y. Cao, E. Khestanova, M. Ben Shalom, A. V. Kretinin, K. S. Novoselov, L. Eaves, I. V. Grigorieva, L. A. Ponomarenko, V. I. Fal'ko, and A. K. Geim, *Science* **357**, 181 (2017).

[Redacted]

(b) K. Komatsu, Y. Morita, E. Watanabe, D. Tsuya, K. Watanabe, T. Taniguchi, and S. Moriyama, *Sci. Adv.* **4**, eaaq0194 (2018).

[Redacted]

Figure R2:

[Redacted]

and hole. So it should be observable from conventional resistance measurement. However, the signal can be reasonably small in both longitudinal resistance and Hall resistance measurement. The resistance measurement always contains large signals originating from main-and sub-Dirac points since they are low carrier densities. In contrast, signals around vHSs are high carrier density thus low resistance and small signal. So that the signal from vHS can be easily hidden by the large signals from DPs. The thermoelectric effect is more sensitive to the low resistance region since it is proportional to the derivative of the conductivity. So, the thermoelectric effect makes it easier to detect signals from vHSs.

Regarding the second question “What other methods exist to measure the valley-g-factor and how do they compare to the one presented here? The most closely related work is Ref.14. Also here a more explicit comparison is needed (e.g. I think that the method of Ref.14 is more accurate as it can be employed at base temperature, but it requires low densities (not possible at the 2nd VHS) and more complex geometries). ”

There can be several different ways to evaluate valley g-factor. The Ref. 14” Tunable Valley Splitting due to Topological Orbital Magnetic Moment in Bilayer Graphene Quantum Point Contacts.” is one example. Such type of energy-dependent electrical conductance measurement can be used for determining valley g factor under the application of a magnetic field. We think not only the point contact but also tunneling spectroscopy [9] or capacitance [6] measurements could be used to detect the magnetic field-induced splitting of vHSs since these measurements allows us to obtain the information about the density of states (DOS) of graphene. Reviewer is right that point contact measurement works better in low carrier density regions such as around charge neutrality point (Dirac point: DP) or near the bandgap; we infer quantized conductance of point contact is difficult to achieve at vHS points. Other methods (tunneling or capacitance) could be used to detect vHS. These are interesting experiments. However, as far as we know there is no published paper for detecting 2nd hole-side vHS with these methods.

We also would like to point out that optical measurement can be also used for determining valley g factor. For transition metal dichalcogenides (TMD) semiconductors, the valley-g-factor was measured by optical measurement [such as photoluminescence (PL) and absorption spectra] under a magnetic field. We believe there have been many pioneering works on this subject as overviewed in our Ref. 1. We also find there is another good review of these works entitled “Valleytronics in 2D materials” written by Schaibley et al. [10]. We include this new reference in the revised manuscript and add comments on the progress on TMD materials with regards to the valley dependent optical properties.

The optical detection scheme used in TMD materials might be used for graphene/h-BN moiré systems if there is an appropriate optical transition that can be used for detecting vHSs. As far as we know, the optical transition between vHSs is studied systematically in the twisted bilayer graphene such as the paper “Chiral atomically thin films” written by Kim et al. [11]. So optical spectroscopy might be used to provide such information. We infer that the energy separation between main and sub DPs, as well as vHS, appear in the energy scale of infrared light, the infrared optical setup in a magnetic field is necessary for this.

Comment 3)

A concluding section with the outlook on twisted bi- and multilayer graphene structure would be interesting to the readers. What kind of features in the complex band structures of these materials could be potentially detected with this setup?

Reply for comment 3)

Thanks for your comment. We agree that experiments on twisted bi- and multilayer graphene would be an interesting subject for magneto thermoelectric measurement. Since valley orbital moment can be nonzero in the system with broken inversion symmetry, our method can be a good probing method for the inversion asymmetry of the bands in twisted graphene. It will be interesting to see if the valley orbital moment is different among different band gaps or vHSs. These discussions are included in the revised manuscript.

Comment 4)

Overall, the manuscript is carefully written and well-structured. The science is accurately described and presented in an understandable way.

Reply for comment 4)

Thanks for your positive comment on our manuscript. We believe the finding of the orbital angular moment at vHS deepens the understanding of the relationship between band structure (topological property) and orbital moment of the wave function. Since orbital moments and Berry curvature have strong connections in between such that if one is non-zero another is also non-zero, these results are containing topics for broad interest.

Comment 5)

I have a few questions regarding the scientific aspects:

- The authors induce a temperature gradient by heating a close-by graphene sheet. Is it possible to provide quantitative simulations? How does heat transfer work through SiO₂? How important is it that the hBN is cut in the middle?

Reply for comment 5)

Thanks for your comments. Regarding the first question “The authors induce a temperature gradient by heating a close-by graphene sheet. Is it possible to provide quantitative simulations?”, we have estimated heat transfer in the device from two different methods shown below. We think these data provide some insight for understanding the heat transfer property in the device.

[Redacted]

Reply to Reviewer 2

[Redacted]

Reply to Reviewer 2

[Redacted]

[Redacted]

Regarding the third question “? How important is it that the hBN is cut in the middle?”, the main purpose of the making cut in the middle of h-BN in our device is to reduce capacitive coupling between the two graphene flakes during the thermoelectric measurement as we illustrated in Figure R7. We saw such an influence of crosstalk particularly for the operation of the graphite gate. We had several optimizations for our device structure particularly for reducing such an effect. The more distance graphite gate to other graphene and cutting in the middle of h-BN does help to reduce electrical cross-talk. The drawback of introducing distance between two graphene devices is reducing heating efficiency from heater graphene to another graphene. We need more heater power to observe the thermoelectric signal from another graphene in the device that has longer separation. The structure shown in our manuscript was optimized for minimizing capacitive coupling and at the same time having reasonable heating efficiency. The cutting the h-BN in the middle seemed to be a small influence for heat transfer from one graphene to another graphene. So the effect of cutting h-BN in the middle is mainly for reducing electrical noise during measurement.

Figure R7: Illustration of the influence of the cutting h-BN for both heat and charge transport.

Comment 6)

- If a power of 1mW is applied, how stable is the temperature? Is there some equilibration time needed before measuring?

Reply for comment 6)

First, our measurement is DC measurement and reasonably slow such that each data point in Fig. 2(e) or Fig. 3(a) in the main text takes about 0.5 seconds to measure (single trace takes ~20 min to measure). In this time scale, we have not noticed any signature of equilibration time during measurement. During measurement, we always monitored how stable the temperature of the heat graphene by monitoring its resistance during the application of current for Joule heating. We did not find significant fluctuation of the resistance and we did not see any sign of gradual changes toward equilibration. We see a similar trend for the graphene in that the temperature gradient is induced. So at least we would think that these devices are stable in the time scale of the above-mentioned, typical time scale for resistance measurement. These suggest that the equilibration time of the device is at least less than a second. We restricted our injection power of 1 mW in maximum and in such a case we have not seen any degradation of device quality.

Application of large power into the graphene has been studied for the application of black body radiation devices up to 20-90 mW using *h*-BN/graphene/*h*-BN devices [14,15]. These literatures showed that graphene is quite stable under the application of large power. We are stayed injecting power far below these literatures, therefore we think our devices should have good long term stability.

Comment 7)

- At $B=0$, there is a wiggle at $nL=-4n_0$ and a small one at $+4n_0$. What is the origin of this signal?

Reply for comment 7)

Thanks for commenting on this. These are the signature of the Seebeck effect of the graphene. Although our main thermoelectric signal is due to the Nernst effect, there is a small contribution from Seebeck effect. The overall difference between the two effects for monolayer graphene is schematically illustrated in Figure R8(a) and R8(b). The difference between Nernst and Seebeck contribution is whether the thermoelectric voltage is

Figure R8: (a,b) Relation among temperature gradient, thermoelectric signal at $B = 0$ and $B \neq 0$ for (a) Nernst and (b) Seebeck effect. (c) Illustration of temperature gradient expected to present in our device.

measured perpendicular to the temperature gradient (Nernst effect) or measured parallel to the temperature gradient (Seebeck effect). The Nernst component of magneto-thermopower is zero at $B = 0$ and becomes finite for $B \neq 0$, while the Seebeck component of thermopower is finite for both 0 and $B \neq 0$. The oscillating change of the thermoelectric signal at $B \neq 0$ is due to the Landau quantization. The Seebeck component of the thermopower induces a signal at the main and sub-Dirac point. Such a contribution from the Seebeck effect can exist in our device as we depicted in Figure R8(c). Although the main temperature gradient is perpendicular to the channel, there is a temperature gradient between the graphene channel and electrode such as graphene is hot, and the electrode is cold; this is because the electrodes are always kept cold. The thermopower originated from this temperature gradient is parallel to the voltage probe direction, thus Seebeck effect can be generated. In principle, the Seebeck contribution can be small since the Seebeck signal at two electrodes have opposite sign so that they tend to cancel each other. However, any asymmetry between two electrodes can make Seebeck signal to be non-zero.

Comment 8)

- There is the statement on page 5 that the valley-g-factor is “unusually” large. How does this $g \sim 140$ compare to other works? Has a valley-g-factor on Moiré/hBN ever been measured before?

Reply for comment 8)

Thanks for pointing out this. As far as we know, there was no research investigated valley g factor at vHSs in graphene/h-BN moire system. We think the sentence the reviewer pointed out is rather an ambiguous statement in the previous version of our manuscript. Here, we wanted to state that valley g factor at hole-side 2nd vHS is very large compared to other vHSs (electron side 1st, hole side 1st vHSs) investigated in our experiment. The comparison was between different vHS in the graphene/h-BN system. In the revised manuscript, we corrected this point by removing the sentence “unusually large” and revised text around here.

Comment 9)

- Please state explicitly how you distinguish the signal at the 2nd VHS from Landau-levels (Fig.3a)

Reply for comment 9)

This is a good point, and this is our motivation for using the Nernst effect for the measurement. The overall behavior of the Nernst effect is illustrated in Figure R8(a). The Nernst effect exhibits sign reversal between $+B$ and $-B$; this is because of the reversal of Lorentz force between different B direction. This sign change is also exhibited when the system is under Landau quantization. In addition to this, there is a sign change of the Nernst effect between the Dirac point (DP) and vHS. This is because DP gives the majority carrier change from electron to hole when one looked from the higher energy side to lower energy. In contrast, at vHS majority carrier changes from hole (higher energy) to electron (lower energy). These difference induces opposite sign of Nernst signal between DP (as well as LLs originated from the DP) and vHS point [12,13,16]. Therefore, the B dependence of the Nernst signal can be used to distinguish signal originated from DP or vHS. An example of this is shown in Figure R9. We plot the Nernst voltage V_{ind} as a function of B for four different DPs. These are the main Dirac point (DP), Secondary Dirac point (SDP), and Third (or tertiary) Dirac point (TDP). The signal from these points tends to increase with B while the signal from all the vHS decrease with B . Thus, there is an apparent difference in the sign of the signals. This information is used to separate the signal from DP. Since we think these discussions are quite important for supporting our conclusion, we included these in the supplementary information in the revised manuscript.

Figure R9: (a) Two-terminal resistance R of the graphene on the left as a function of its carrier density n_L at 2.0 K. (b) V_{ind} as a function of the carrier density n_L measured at a different out-of-plane magnetic field B obtained from the graphene/ h -BN moiré superlattice device at 2.0 K. The dotted black lines depict the carrier density of the DP, SDP(e), and SDP(h). The red, blue, and green lines depict 2nd vHS(h), 1st vHS(h), and 1st vHS(e), respectively. Traces are offset for clarity and offset is depicted by black dashed lines. (c,d,e,f) V_{ind} as a function of the B for four DPs of (c) TDP(h), (d) SDP(h), (e) DP, and (f) SDP(e). (g,h,i) V_{ind} as a function of the B for three vHSs of (g) 2nd vHS(h), (h) 1st vHS(h), and (i) 1st vHS(e).

The signals of LLs are having the same nature with the DP. In Figure R10, we show a simple example of the Nernst effect of the h-BN/mono-layer graphene/h-BN structure. The signal from LLs tends to be positive for $+B$ and negative for $-B$ as the signal from DPs are also positive for $+B$ and negative for $-B$. So, once we assigned the DP from low field signals, the LLs originated from the DP can be easily tracked.

Figure R10: (a) Schematic illustration of thermoelectric voltage detection in a parallel graphene device. (b) Two-terminal resistance R of the nonaligned graphene on the right as a function of its carrier density n_R at 2.0 K. (c) Thermoelectric voltage V_{ind} generated in the nonaligned graphene as a function of its carrier density n_R , measured at a different out-of-plane magnetic field $B = \pm 1.8, \pm 0.9$, and 0 T. A constant power P of 1 mW is applied to the aligned graphene on the left. (d) V_{ind} as a function of n_R and B .

Comment 10)

- Page 7: "The signals arising from three vHSs—the first vHS(e), first vHS(h), and second vHS(h)—are extracted from the data, as shown in Fig. 3(b)". How was this extraction done? State explicitly.

Reply for comment 10)

Based on the discussion in the Reply for comment 9), we can distinguish the vHSs from DPs. Then, as you can see from Fig. 2(g) or Fig. 3(g) of the main text, we took the bottom of dip for positive B , and top of the peak for $-B$ region for extracting positions of vHSs. We add this explanation in detail in the revised manuscript.

Comment 11)

- How strongly were the parameters in the continuum model adjusted in order to fit the experimental data?

Reply for comment 11)

This is a good point. First, we would like to remind the reviewer that the continuum model for graphene/h-BN moire superlattice had been investigated in the presence of inversion asymmetry by the co-author of our manuscript (M. Koshino, P. Moon) [3,17]. As reviewer 1 suggested, a similar calculation has been done by another group in which assumption was made such that inversion asymmetry is small [1,2]. The results of these papers are reasonably consistent with each other except the former one showed band gap opening in main-DP and sub-DPs, while later one does not show band gap opening effect. Both of papers provided a

good agreement with experimental observations of Landau fan diagram including the features of Hofstadter butterfly in graphene/h-BN system.

For band calculation, we used exactly the same parameter with ref. [3,17] for lattice misorientation between graphene and h-BN with 0 deg. The calculated band structure using inversion asymmetric model in Fig. 1(d) and 1(e), and theoretical calculation in Fig. 3 are basically identical to the band structure shown in Fig. 3(b) and 3(c) of ref. [3] and Landau level structure shown in Fig. 6(d) of ref. [3], respectively. We basically extend calculation range to larger energy values and re-calculated the low magnetic field region with more fine data points. So, there is no intentional adjustment of the parameter to fit the result presented in our manuscript.

According to ref. [3], the increase of misorientation angle between graphene and h-BN tends to make a separation between secondary DP and vHSs, therefore, the separation of signal from these points can be more difficult in larger misorientation angle (e.g. $\theta = 1\sim 2$ deg.). There is another parameter such that the difference in the influence of potential between two different carbon atoms [C_A and C_B in Fig. 1(a)] due to the h-BN. This can influence the results. Obviously, no inversion asymmetry makes the orbital moment to be zero everywhere in the band. Introduction of difference in C_A and C_B generates band gap opening at main and sub-DPs and generates orbital moment. We think that since the comparison between calculated DOS in Figs. 3(c,d,e) and experimental data Fig. 3(a,b) showed reasonably good agreement with each other such that the change of peak positions in the experiment is reproduced by the calculation, the parameters developed in ref. [3] was already good enough to explain our result even without any extra adjustment. In the revised manuscript, we included sentence that explain that the parameter for inversion asymmetric model is same as previously published ref. [3].

Comment 12)

- Why is the semi-classical model that is used to extract the orbital magnetic moment still valid at large magnetic field, i.e. in the presence of Landau levels?

Reply for comment 12)

The signal we are interested in is presented in the low field range (below ~ 1.5 T) so that Landau levels (LLs) are not that pronounced. In addition to this, since the vHSs are in high carrier density region where the contribution from the LLs are smaller than around DPs since energy spacing between the LLs are small. Therefore, the thermoelectric signals originated from vHSs are not so much influenced by LLs as far as we limit our discussion in the range around zero field and low magnetic field. We plot all the assignments of the magnetic field dependence of DPs and vHSs in Figure R11. In the low field (\sim below 1.5 T), the positions of vHSs are reasonably separated from low-index LLs such as filling factor $\nu = \pm 8$ etc.

Secondary, according to ref. [18,19] written by Xiao, Niu et al., orbital moment behaves exactly like the electron spin. Therefore, in the semiclassical picture, in the presence of a weak magnetic field \mathbf{B} , the electron band structure energy $\varepsilon_0(\mathbf{k})$ acquires a correction term from the intrinsic orbital magnetic moment $\mathbf{m}(\mathbf{k})$ such as $\varepsilon(\mathbf{k}) = \varepsilon_0(\mathbf{k}) - \mathbf{m}(\mathbf{k}) \cdot \mathbf{B}$. Within this range, $\varepsilon_0(\mathbf{k})$ can be regarded as band energy of graphene including Landau quantization; then the second term provides excess energy due to the orbital magnetic moment and it is linear to the \mathbf{B} . As we can see from Fig. 3(b), the splitting of hole-side 2nd vHS [2nd vHS(h)] exhibiting linear change with \mathbf{B} , this is what we expected from the above-mentioned model. Therefore, we think the semiclassical picture is still valid in the low field region we studied.

Figure R11: (a) Image plot of V_{ind} as a function of VBG and B for aligned graphene/h-BN Moiré superlattice device showing fan diagram measured at 2.0 K. (b) Assignment for Landau levels originated from main- and sub-DPs, as well as a assignment for the position of signal from vHSs.

Comment 13) Minor comments

- Check font sizes in Figures

Reply for comment 13)

Font size were small in all the figures, so we changed it to be bigger font in the revised manuscript.

Comment 14) - If traces are offset from each other for clarity, this should be state explicitly

Reply for comment 14)

Trances are offset in Fig. 2(e,f) and Fig. 3(f,g). We add explanation for the offset in the figure caption in the revised manuscript.

Comment 15) - Figure 1fgh: Are these the same colors as in d? It is difficult to link the plots to the points in the band structure and the label is tiny.

Reply for comment 15)

Thanks for your comment. Fig. 1(f,g,h) and Fig. 1(d) is not using same color. We modified the plot and labels in the revised manuscript.

Comment 16) - Do Fig. 2g and 3g show the same data? If yes, that should be stated explicitly.

Reply for comment 16)

Fig. 2(g) and Fig. 3(g) is same data, so we add the comment for this in the revised manuscript in main text as well as figure caption.

Comment 17) - Page 7: “The variations in V_{ind} with respect to n_L measured using ...”. What is the precise meaning of the “variation in V_{ind} ”?

Reply for comment 17)

Thanks for your comment. The corresponding sentence “The variations in V_{ind} with respect to n_L measured using a magnetic field increasing from $B = 0$ T to 2.23 T are shown in Fig. 2(g).” was intended to state that V_{ind} vs. n_L curve is measured under different magnetic field B values and plotted in Fig. 2(g). The reviewer is right that the sentence “variations” is rather unclear at here. We modified this sentence in the revised manuscript. Thanks for pointing out this.

In addition to the above-mentioned replies for reviewer’s comments and changes in the revised manuscript, there are other two corrections in the manuscript that we would like to ask you to check. These corrections have appeared when we consider the question from the Reviewer 3.

[Correction 1]

(Statement in the submitted manuscript) We stated that during magneto-thermoelectric effect measurement, heater graphene’s carrier density is tuned to its charge neutrality (DP) under the application of current for Joule heating.

(Correct statement) During magneto-thermoelectric effect measurement, heater graphene’s carrier density n_R is tuned to the quantum Hall filling factor $\nu = -6$ such that $n_R = 6eB/h$ during the application of current for Joule heating. This means that carrier density at zero magnetic field is charge neutrality $n_R = 0$ same as previous manuscript, however $n_R = 6eB/h$ under the application of magnetic field.

The intension of both methods is to keep heater graphene’s condition to be constant. Keeping graphene’s carrier density to the quantum Hall filling factor $\nu = -6$ keeps graphene heater’s total resistance to be around 10 k Ω (channel resistance + contact resistance). The detail of this can be found in (Reply to comment 1) section for Reviewer 3. We found this mistake when we considered questions from reviewer 3. We deeply apologize for this; this correction was also mentioned to the other two reviewers. We are very glad that if you could check this revised manuscript.

[Correction 2]

We also wanted to correct valley g-factor at the 2nd hole-side vHS from $g \sim 140$ (previous manuscript) to $g \sim 130$ (in the revised manuscript). This is because since during the initial submission of the manuscript, we determined the amplitude of orbital magnetic moment $m(k)$ from the color scale of the contour plot as shown in Figure R12(b,c,d) below. Then we determined $m(k) \sim 70 \mu_B$ and this corresponding to valley g-factor of $g = 2m(k) = 140$. During the consideration of comment from reviewer 3, we noticed that a more precise value of calculated $m(k)$ is $m(k) = 66.4 \mu_B$ as you can see from Figure R12(e,f,g). In Figure R12(e,f,g), we plotted calculated $m(k)$ values along Y-point to X-point in k-space. The $m(k) = 66.4 \mu_B$ could be approximated as $m(k) \sim 70 \mu_B$, however, it is not appropriate to claim that $g \sim 140$ based on this. Since the more precise g-factor value is $g = 2m(k) = 132.8$, we think it is more correct to state that determined valley g-factor is $g \sim 130$. Because of these reasons, we would like to correct $g \sim 140$ to be $g \sim 130$ in the revised manuscript. We believe that this correction does not have a significant influence on the main claim of our manuscript; however, we would like to let all three reviewers know about this change. We are very glad if you could consider this correction in addition to all the replies to your comment.

Figure R12: (a) Band structure of the graphene/h-BN Moiré superlattice with $\theta = 0^\circ$ at K-point calculated using an effective continuum model. Calculation is performed under an inversion asymmetric model. (b,c,d) Calculated orbital angular momentum m for hole-side (b) first band, (c) second band, and (d) third band. (e,f,g) Line profile of m vs. k between Y-point to X-point for (e) first band, (f) second band, and (g) third band.

Reviewer: 3

General comment)

The authors report a large orbital magnetic moment at the second hole van Hove singularity in graphene/hBN moire superlattice. The main evidence is the magnetic field splitting of Nernst effect peaks at vHs, which agrees with model calculation. It provides interesting insight into the band structure of graphene/hBN moire structure.

Reply for general comment)

Thanks for reviewing our manuscript. Please find below the one-by-one answer to your comments.

Comment 1)

Can the authors include more detail about the device while generating heat, e.g. what is the carrier density and channel resistivity? What is the contact resistance? This is relevant because it affects where the hot spot is.

Reply for comment 1)

Thanks for your comment. The two-terminal resistance the heater graphene under the low magnetic field region $B = 0, 1, \text{ and } 2.0 \text{ T}$ is presented in Figure R13(a). First, we extract the contact resistance contribution of the device. At the highest carrier density ($\pm 2.37 \times 10^{12} \text{ cm}^{-2}$), resistance tends to saturate to the lowest value. As a rough estimation of contact resistance, we assumed that the two-terminal resistance at these highest carrier densities is dominated by the two metal/graphene contact resistance. Then we obtained $\sim 5.0 \text{ k}\Omega$ and $\sim 3.4 \text{ k}\Omega$ for hole-doped ($n_R = -2.37 \times 10^{12} \text{ cm}^{-2}$) and electron-doped ($n_R = +2.37 \times 10^{12} \text{ cm}^{-2}$) side. Next contact resistance can be estimated from the deviation from quantum Hall resistance value at the high magnetic field as shown in Figure R13(b). The quantum Hall resistance at filling factor $\nu = \pm 2$ and ± 6 are $12906.4 \text{ }\Omega$ and $4302.13 \text{ }\Omega$, respectively (these values are indicated by the dashed line in the Figure R13(b)). The difference between these values and measured two-terminal resistance can be attributed to the contact resistance contribution. Obtained values at $\nu = \pm 2$ and ± 6 are plotted as filled red square in Figure R13(c) together with the contact resistance extracted by first method (filled black square). Both measurements provide similar values for contact resistance of $\sim 5 \text{ k}\Omega$ and $\sim 3 \text{ k}\Omega$ for the hole- and electron-doped side, respectively. During the thermopower measurement with applying the power of 1 mW to heater graphene, we adjust gate voltage to

Figure R13: (a) Two-terminal resistance R of a heater graphene as a function of carrier density measured under different magnetic field B measured at 2.0 K . (b) R vs. n at $B = 8.7 \text{ T}$. Quantum Hall plateaus of filling factor $\nu = -6, -2, 2, \text{ and } 6$ are clearly visible. (c) Total contact resistance contribution of the Au/Cr/graphene junction (filled red square) determined for different n values. The contact resistance contribution determined from the saturation of panel (a) is plotted as filled black square.

tune the carrier density of heater graphene to be at $\nu = -6$ such that $n_R = 6eB/h$, where e depicts the electron charge and h the Planck constant. So two-terminal device resistance is kept around $10 \text{ k}\Omega$ (Figure). Note that two-terminal device resistance at DP is also $\sim 10 \text{ k}\Omega$. Therefore, device resistance (contact resistance + graphene's channel resistance) is always kept $\sim 10 \text{ k}\Omega$ from zero to the high magnetic field. In this way, we achieved constant resistance of the heater graphene throughout the measurement. Since the dimension of the heater graphene channel is $9.4 \text{ }\mu\text{m}$ (length) and $3.4 \text{ }\mu\text{m}$ (width), the sheet resistance of the graphene channel is $\sim 1.5 \text{ k}\Omega/\square$. From these comparisons, the contact resistance and channel resistance are in a similar order. Both contact resistance and channel resistance are much higher than the resistance of Au/Cr electrode. Therefore, we think at least heat generation is localized around the heater graphene device (heater graphene channel and interface between heater graphene and metal electrode). Experimentally, it is difficult to determine whether there is a hot spot and if it exists, where is the hot spot in current our measurement setup. However, we would like to draw the reviewer's attention that the thermoelectric effect contains two different contributions, Nernst and Seebeck effect as shown in the figure. Any inhomogeneous heating will generate the Seebeck effect that is the almost same order of thermoelectric coefficient with the Nernst effect, but different in the shape of the signal; Seebeck effect exists at zero magnetic fields. According to our figure, the signal at zero magnetic field is much smaller than the signal in the field. This is a piece of evidence that our structure provides homogeneous heating to moiré graphene to exhibits dominant contribution from the Nernst effect. We included these discussions in the revised manuscript.

Figure R14: (a,b) Relation among temperature gradient, thermoelectric signal at $B = 0$ and $B \neq 0$ for (a) Nernst and (b) Seebeck effect.

During this correction that we noticed that there was a mistake in the submitted version of the manuscript as follows;

(Statement in the submitted manuscript) We stated that heater graphene's carrier density is tuned to its charge neutrality (DP) under the application of current for Joule heating.

(Correct statement) Heater graphene's carrier density n_R is tuned to the quantum Hall filling factor $\nu = -6$ such that $n_R = 6eB/h$ during the application of current for Joule heating. This means that carrier density at zero magnetic field is charge neutrality $n_R = 0$ same as previous manuscript, however $n_R = 6eB/h$ under the application of magnetic field.

We deeply apologize for this mistake; this correction was also mentioned to the other two reviewers. We are very glad that if you could check this revised manuscript.

Comment 2)

The authors mentioned that "The opening of a large gap with inversion asymmetry naturally results in the generation of an orbital magnetic moment $m(\mathbf{k})$ at this (second) vHS". Is the main Dirac point also gapped within the inversion asymmetry model? If yes, is it understood why a similar effect reported for the second vHS is not observed for the first hole and electron side vHS? Would it be possible to provide an intuitive picture of what leads to the exceptionally large orbital magnetic moment in the second hole vHS?

Reply for comment 2)

Thanks for asking this important question. The reviewer is correct that the main Dirac point also has a small bandgap within the inversion asymmetry model. The size of the gap in the calculation presented in Fig. 1(d) is ~ 2 meV.

Regarding the question "is it understood why a similar effect reported for the second vHS is not observed for the first hole and electron side vHS?". We would like to draw your attention to the semiclassical model for describing orbital magnetic moments that is given by the following expression as it is shown in the main text of the manuscript.

$$\mathbf{m}(\mathbf{k}) = -i \frac{e}{2\hbar} \langle \nabla_{\mathbf{k}} u | \times [H(\mathbf{k}) - \varepsilon^0(\mathbf{k})] | \nabla_{\mathbf{k}} u \rangle \quad (1)$$

This equation can be rewritten into the following form [20-22];

$$\mathbf{m}_n(\mathbf{k}) = -i \frac{e}{2\hbar} \sum_{i \neq n} \frac{P_{n,j}(\mathbf{k}) \times P_{j,n}(\mathbf{k})}{\varepsilon_j^0(\mathbf{k}) - \varepsilon_n^0(\mathbf{k})} \quad (2)$$

Here $P_{n,j}(\mathbf{k}) \equiv \langle u_{n,\mathbf{k}} | \hat{\mathbf{p}} | u_{j,\mathbf{k}} \rangle$ is the interband matrix element of the canonical momentum operator $\hat{\mathbf{p}}$. $\varepsilon_n^0(\mathbf{k})$ is the dispersion of n th band and $\varepsilon_n(\mathbf{k}) = \varepsilon_n^0(\mathbf{k}) - \mathbf{m}_n(\mathbf{k}) \cdot \mathbf{B}$ is the electron energy. To obtain a finite orbital moment, the both numerator and denominator need to be nonzero in eq. (2). Since the denominator of eq. (2) corresponds to the energy separation to neighboring bands, it is mostly nonzero everywhere in the band except the point of contact between the bands such as Dirac point (zero-gap Dirac point).

Then, the numerator, interband matrix element is crucial for obtaining large $\mathbf{m}(\mathbf{k})$. In an inversion symmetric model, this value is finite only at the point where two bands make contact as indicated by red arrows in the Figure R15. The interband matrix elements at this point will be conserved to be finite even after the gap opening under to the introduction of inversion asymmetry. Below, we present a few examples of this. Point **A** in Figure R15(a) is the one we think is most interesting. This point is the band touching point in the symmetric model thus the numerator of eq. (2) is finite. In the asymmetric model [Figure R15(b)], a large gap is opened at his point and generate 2nd vHS(h) in the second band. So, this point satisfies the criteria of the finite interband matrix element and since there is a finite gap between the neighboring band; thus, point **A** can exhibit a large orbital moment [please see Figure 1(f,g,h)].

Comparing with this, now we discuss point **B** in Figure R15(a). This point is not the point of contact between the bands, but a vHS of 1st band in the symmetric model. Thus numerator, interband matrix element is zero. In the asymmetric model, it is still vHS of the 1st band. Introduction of inversion asymmetry slightly modify the band structure to generate finite orbital moment $\mathbf{m}(\mathbf{k})$, however since the numerator, interband matrix is kept small value even after the introduction of inversion asymmetry, obtained $\mathbf{m}(\mathbf{k})$ is small at point **B** [please see Figure 1(f,g,h)].

Other points of contact between the bands in the symmetric model (point **C**, **D**, and **E**) will be gaped in the inversion asymmetric model and become main- and secondary-DP; these points have a large orbital moment as shown in Figure 1(f,g,h) and Supplementary Figure 2. Overall, point of contact between the band in the symmetric model always shows large orbital moment when this point is gapped under inversion asymmetric model. These explain our experimental observation of large orbital moment at 2nd vHS(h).

Above-mentioned discussions are included in the supplementary information of the revised manuscript.

Figure R15: (a,b) Band structure of the graphene/h-BN moiré superlattice with $\theta = 0^\circ$ at K-point calculated using an effective continuum model. Calculation is performed under an (a) inversion symmetric model and an (b) inversion asymmetric model.

Comment 3)

In figure 1d, what is causing the slope background in the resistance? In figure 1e, V_{ind} seems to diverge on the left end of the plot, what is the reason for that? In figure 3g, the V_{ind} signal for the right red dot is always stronger than the left dot, does this indicate one valley has larger DOS than the other? Can the authors comment on this?

Reply for comment 3)

[1] We believe that the first question of the reviewer “In figure 1d, what is causing the slope background in the resistance?” is regarding the slope background in the resistance that appeared in Fig. 2(d). We infer this originated from the inhomogeneity of metal/graphene contact. As it can be seen from the Figure R16, two-terminal resistance at the highest electron (hole) density is $\sim 2 \text{ k}\Omega$ ($\sim 6 \text{ k}\Omega$) and this value is mostly the contact

Figure R16: Two-terminal resistance of moiré graphene. Illustration is showing effect of doping around the contact.

resistance values of Au/Cr/graphene junction. If this is contact resistance, we think it is not the lowest contact resistance value in literature; so that there is some imperfection there. In our experience, Cr/graphene interface induces n-type doping in the graphene side, so in the presence of an n-doped graphene region near the contact, there can be a n/n junction around metal contact when the graphene channel is n-doped. Further, when graphene is p-doped, there can be a p/n junction of graphene around the metal contact. Thus, in the presence of n-doped graphene regions around the contact, the two-terminal resistance of the device can be larger in the hole-doped region due to the pn junction. In the presence of some inhomogeneity around the contact such that inhomogeneous n-doping etc., we infer that there can be such gradual changes in the two-terminal resistance. We add comment on this in the revised manuscript.

[2] For the second question “In figure 1e, V_{ind} seems to diverge on the left end of the plot, what is the reason for that?”, we believe that it is regarding the increase of V_{ind} in Fig. 2(e) indicated by red arrow in Figure R17(a) below. Figure R17(a) shown below is same figure as Fig. 2(e) in the main text. For better visibility, we presented Figure R17(a) with different Y-axis. Thanks for asking this, this is an interesting point and we did not comment on the submitted version of the manuscript. Comparing Figure R17(a) with Figure R17(b), we think that the position of divergence seems to be coincident with the 3rd Dirac point (Tertiary Dirac point: TDP) as indicated by the black open square in Figure R17(b).

As another comparison, a magnetic field dependence of the Nernst signal from this point is plotted in Figure R18, comparing with the magnetic field dependence on other DPS and vHSs. The signal is positively (negatively) increases for $+B(-B)$ value similar to other DPs such as main DP (at $n = 0$) and sub-DPs. For these reasons, we think that the diverge at the left end of the plot in Figure R17(a) is due to the Nernst effect on the hole-side TDP. This is interesting that the signal at this Dirac point looks much larger than other features. The hole side TDP is having a more complicated band structure than other DPs as you can see from Figure

Figure R17: (a) Thermoelectric voltage V_{ind} generated in the graphene on the left as a function of its carrier density n_L , measured at a different out-of-plane magnetic field $B = \pm 0.45, \pm 0.22$, and 0 T. A constant power P of 1 mW is applied to the graphene on the right. Traces are offset for clarity and offset is depicted by black dashed lines. (b) Calculated DOS with respect to normalized carrier density n/n_0 . (c) Band structure of the graphene/h-BN moiré superlattice with $\theta = 0^\circ$ at K-point calculated using an effective continuum model. Calculation is performed under an inversion asymmetric model.

R17(c); it is not exactly a Dirac-like band anymore, the lowest energy of 2nd band is located at K-point, while the highest energy of 3rd band is at Y-point. They are not in the same location. In addition to this, this point has band inversion (Y-point at 3rd band is located in higher energy than K-point of 2nd band), so it is a semi-metallic character. Some of these unique properties of TDP might give rise to show a large Nernst signal, yet we think it is necessary to investigate more detail.

The gate voltage required to reach this point is almost the limit of h -BN/SiO₂ gate dielectric; we could not apply further negative values of gate voltage so that we cannot really investigate much detail on this point. We are interested in investigating this in future experiments. The use of h -BN instead of h -BN/SiO₂ as a gate dielectric could be used to investigate such high carrier density regime. The diverge on the V_{ind} on the left end of the plot was commented in the revised manuscript with the explanation that this is originated from TDP.

Figure R18: (a) Two-terminal resistance R of the graphene on the left as a function of its carrier density n_L at 2.0 K. (b) V_{ind} as a function of the carrier density n_L measured at a different out-of-plane magnetic field B obtained from the graphene/ h -BN moiré superlattice device at 2.0 K. The dotted black lines depict the carrier density of the DP, SDP(e), and SDP(h). The red, blue, and green lines depict 2nd vHS(h), 1st vHS(h), and 1st vHS(e), respectively. Traces are offset for clarity and offset is depicted by black dashed lines. (c,d,e,f) V_{ind} as a function of the B for four DPs of (c) TDP(h), (d) SDP(h), (e) DP, and (f) SDP(e). (g,h,i) V_{ind} as a function of the B for three vHSs of (g) 2nd vHS(h), (h) 1st vHS(h), and (i) 1st vHS(e).

[3] For the third question, “In figure 3g, the V_{ind} signal for the right red dot is always stronger than the left dot, does this indicate one valley has larger DOS than the other? ”. Please see the figure below for the V_{ind} signal for the application of both $+B$ and $-B$ directions in Figure R19. In the original Fig. 3(g), we only showed data for $+B$. Here we show both $+B$ and $-B$ direction. The reversal of the magnetic field direction changes the relation between K and K' valleys such that K' valleys move to low carrier density with increasing $+B$ value, while the same valley moves to higher carrier density with increasing $-B$ value. This is since the valley splitting is caused by the orbital magnetic moment that behaves similarly to the magnetic moment. In both magnetic field directions, we still see that the peak at the low carrier density side is exhibiting larger signals than the peak on the high hole density side. So, we would think it is not because one valley has larger DOS than another valley. The thermoelectric signal is, in principle, proportional to the $d\sigma/dE$ rather than the DOS itself. So, we infer that it should be originated from the difference in the $d\sigma/dE$ between the two valleys. Roughly $d\sigma/dE$ proportional to the energy derivative of DOS rather than DOS itself. So, it could be since the energy-dependent shape of DOS can be different between two valleys and that cause asymmetry in the signal. However, at this moment, we do not have a clear idea of explaining this feature.

Figure R19: V_{ind} as a function of the carrier density n_L measured at a different out-of-plane magnetic field B obtained from the graphene/I-BN moiré superlattice at 2.0 K. Peak positions of V_{ind} corresponding to K and K' valley are depicted by open red square and filled red circle, respectively. Traces are offset for clarity and offset is depicted by black dashed lines.

Comment 4)

Can the authors show in a separate plot either in the SI or main text, another version of Figure 3 c-e with horizontal axis being energy, with which the actual $m(\mathbf{k})$ value can be more easily extracted? I also recommend including the details of how to convert energy to density in the calculation.

Reply for comment 4)

Please find the data for Fig. 3(c,d,e) with the horizontal axis being energy in Figure R20 shown below. This is included in the SI in the revised manuscript. From the slope of splitting energy between the two valleys, valley g-factor g is calculated from the equation, $g = \Delta E / \mu_B B$ where μ_B denote the Bohr magneton. From this analysis, we obtained $g \sim 116$. From the semiclassical calculation, we obtained $m(\mathbf{k}) \sim 66 \mu_B$ at the 2nd vHS(h) as shown in Fig. 1(g) of main text or Figure R21; this corresponds to the g -factor of $g = 2 m(\mathbf{k}) \sim 132$. So, these two values are reasonably consistent each other.

Figure R20: (a,b,c) Calculated DOS as a function of the energy E and B under the condition of (a) total DOS including both K and K' valley, (b) DOS of K valley, and (c) DOS of K' valley. (d) DOS with respect to the energy E calculated at different magnetic field values. Traces are offset for clarity and offset is depicted by black dashed lines. (e) The peak positions of DOS for 2nd vHS(h) versus different magnetic field B . (f) The energy splitting ΔE versus B .

During the consideration of your comment, we noticed that we need to correct valley g-factor at the 2nd hole-side vHS from $g \sim 140$ (previous manuscript) to $g \sim 130$ (in the revised manuscript). This is because since during the initial submission of the manuscript, we determined the amplitude of orbital magnetic moment $m(\mathbf{k})$ from the color scale of the contour plot as shown in Figure R21(b,c,d). Then we determined $m(\mathbf{k}) \sim 70 \mu_B$ and this corresponding to valley g-factor of $g = 2m(\mathbf{k}) = 140$. During the consideration of your comment, we noticed that a more precise value of calculated $m(\mathbf{k})$ is $m(\mathbf{k}) = 66.4 \mu_B$ as you can see from Figure R21(e,f,g). In Figure R21(e,f,g), we plotted calculated $m(\mathbf{k})$ values along Y-point to X-point in k-space. The $m(\mathbf{k}) = 66.4 \mu_B$ could be approximated as $m(\mathbf{k}) \sim 70 \mu_B$, however, it is not appropriate to claim that $g \sim 140$ based on this. Since the more precise g-factor value is $g = 2m(\mathbf{k}) = 132.8$, we think it is more correct to state that determined valley g-factor is $g \sim 130$. Because of these reasons, we would like to correct $g \sim 140$ to be $g \sim 130$ in the revised

manuscript. We believe that this correction does not have a significant influence on the main claim of our manuscript; however, we would like to let all three reviewers know about this change. We are very glad if you could consider this correction in addition to all the replies to your comment.

Figure R21: (a) Band structure of the graphene/h-BN Moiré superlattice with $\theta = 0^\circ$ at K-point calculated using an effective continuum model. Calculation is performed under an inversion asymmetric model. (b,c,d) Calculated orbital angular momentum m for hole-side (b) first band, (c) second band, and (d) third band. (e,f,g) Line profile of m vs. k between Y-point to X-point for (e) first band, (f) second band, and (g) third band.

The answer for second question, “I also recommend including the details of how to convert energy to density in the calculation.” is following. Since what we are calculating is the relation between DOS and energy (figure xx and figure in the main text), we have absolute value of calculated DOS. the carrier density can be calculated by integrating DOS such that $n(E_F) = \int_0^{E_F} \text{DOS}(E)dE$. This is included in the revised manuscript.

Comment 5)

Can the authors explain the inclusion of the inversion-symmetric model in the manuscript? Previous experiments, as cited in this manuscript, already point to the inversion-asymmetric models.

Reply for comment 5)

Thanks for pointing out this. Although the previous experiments showed that inversion asymmetric model provided better coincident with the experimental result, we presented this figure to emphasis the fact that inclusion of inversion asymmetry induces significant changes on band character at second hole-side vHS. The influence of inversion asymmetry on other vHSs are small as you can see the comparison of both symmetric and asymmetric model. This figure is also necessary to explain why this vHS has large orbital moment while other vHS showed small orbital moment contribution. Please refer our answer for your comment 2) section. In the absence of symmetric model intuitive explanation of large orbital moment contribution at second hole vHS is difficult. We add more detail explanation for emergence of 2nd hole-side vHS in the revised manuscript. In such a case, we think presence of inversion symmetric model is more important.

Comment 6)

Typo. The letter “m” in “moire” should not be in caps.

Reply for comment 6)

The reviewer is correct that all the “Moiré” need to be replaced with “moiré”. We corrected this in the revised version of our manuscript. Thanks for finding this error.

References

- 1 Wallbank, J. R., Mucha-Kruczyński, M., Chen, X. & Fal'ko, V. I. Moiré superlattice effects in graphene/boron-nitride van der Waals heterostructures. *Ann. Phys.* **527**, 359-376, (2015).
- 2 Wallbank, J. R., Patel, A. A., Mucha-Kruczyński, M., Geim, A. K. & Fal'ko, V. I. Generic miniband structure of graphene on a hexagonal substrate. *Phys. Rev. B* **87**, 245408, (2013).
- 3 Moon, P. & Koshino, M. Electronic properties of graphene/hexagonal-boron-nitride moiré superlattice. *Phys. Rev. B* **90**, 155406, (2014).
- 4 Komatsu, K. *et al.* Observation of the quantum valley Hall state in ballistic graphene superlattices. *Sci. Adv.* **4**, eaaq0194, (2018).
- 5 Krishna Kumar, R. *et al.* High-temperature quantum oscillations caused by recurring Bloch states in graphene superlattices. *Science* **357**, 181-184, (2017).
- 6 Yu, G. L. *et al.* Hierarchy of Hofstadter states and replica quantum Hall ferromagnetism in graphene superlattices. *Nat. Phys.* **10**, 525-529, (2014).
- 7 Hunt, B. *et al.* Massive Dirac fermions and Hofstadter butterfly in a van der Waals heterostructure. *Science* **340**, 1427-1430, (2013).
- 8 Wang, L. *et al.* Evidence for a fractional fractal quantum Hall effect in graphene superlattices. *Science* **350**, 1231-1234, (2015).
- 9 Kim, H. *et al.* Accurate Gap Determination in Monolayer and Bilayer Graphene/h-BN Moiré Superlattices. *Nano Lett.* **18**, 7732-7741, (2018).
- 10 Schaibley, J. R. *et al.* Valleytronics in 2D materials. *Nat. Rev. Mater.* **1**, 16055, (2016).
- 11 Kim, C.-J. *et al.* Chiral atomically thin films. *Nat. Nanotechnol.* **11**, 520-524, (2016).
- 12 Zuev, Y. M., Chang, W. & Kim, P. Thermoelectric and magnetothermoelectric transport measurements of graphene. *Phys. Rev. Lett.* **102**, 096807, (2009).
- 13 Checkelsky, J. G. & Ong, N. P. Thermopower and Nernst effect in graphene in a magnetic field. *Phys. Rev. B* **80**, 081413, (2009).
- 14 Son, S.-K. *et al.* Graphene hot-electron light bulb: incandescence from hBN-encapsulated graphene in air. *2D Mater.* **5**, 011006, (2017).
- 15 Kim, Y. D. *et al.* Ultrafast Graphene Light Emitters. *Nano Lett.* **18**, 934-940, (2018).
- 16 Kinoshita, K. *et al.* Photo-Nernst detection of cyclotron resonance in partially irradiated graphene. *Appl. Phys. Lett.* **115**, 153102, (2019).
- 17 Koshino, M. & Moon, P. Electronic Properties of Incommensurate Atomic Layers. *J. Phys. Soc. Jpn.* **84**, 121001, (2015).
- 18 Xiao, D., Shi, J. & Niu, Q. Berry Phase Correction to Electron Density of States in Solids. *Phys. Rev. Lett.* **95**, 137204, (2005).
- 19 Xiao, D., Chang, M.-C. & Niu, Q. Berry phase effects on electronic properties. *Rev. Mod. Phys.* **82**, 1959–2007, (2010).
- 20 Chang, M.-C. & Niu, Q. Berry phase, hyperorbits, and the Hofstadter spectrum: Semiclassical dynamics in magnetic Bloch bands. *Phys. Rev. B* **53**, 7010-7023, (1996).
- 21 Yao, W., Xiao, D. & Niu, Q. Valley-dependent optoelectronics from inversion symmetry breaking. *Phys. Rev. B* **77**, 235406, (2008).
- 22 Xu, X., Yao, W., Xiao, D. & Heinz, T. F. Spin and pseudospins in layered transition metal dichalcogenides. *Nat. Phys.* **10**, 343–350, (2014).

REVIEWERS' COMMENTS:

Reviewer #1 (Remarks to the Author):

The manuscript has been revised according to the referees comments, and all questions of the referees have been answered convincingly. I recommend the publication of this manuscript in its current form.

Reviewer #2 (Remarks to the Author):

The authors have revised a few key points in the second version of the manuscript and very thoroughly answered the question that I raised. The paper is publishable in NC and only minor comments need to be addressed.

It would be very beneficial for the reader, if a bit more of the answer to the reviewers were included in the manuscript in a concise form. Especially, this concerns the answer to my second comment, showing that the 2nd VHS at the hole side is also present in the paper by Kumar and the paper by Komatsu, even though it is not properly discussed there. To properly account for this, I suggest to adjust the following sentences:

"In this study, we observed that there was another point (i.e., a second vHS in the holeside located at the Y-point [red-dashed circle in Fig. 1(d)]) that exhibited a pronounced effect of inversion asymmetry; this point has not been experimentally investigated to date."

to:

"...of inversion asymmetry; this point has not been discussed to date."

and:

"The first vHSs on both the electron- and hole-side had previously been investigated [25]; here, we investigated the second vHS(h) which was not studied before."

to :

"...second vHS(h) which was observed [6,7] but not discussed before."

and:

"in Fig. 2(g). The dip-shape signal generated from the second vHS(h) exhibits significant splitting with increasing B, ..."

to:

"in Fig. 2(g). The dip-shape signal generated from the second vHS(h) exhibits significant splitting with increasing B (see also Refs. [6,7]), ..."

Concerning the answers to my comment on the temperature gradient (comment 5): The given answer is clear and detailed and I agree with the authors that it makes sense to publish this discussion in a separate paper. If this paper already exists, it would be nice to have the reference included in the revised manuscript.

On the question on how important the etching of the hBN is: Please include the discussion and Figure R7 to the supporting information.

Finally, the answer to comment 7 on the wiggle at $B=0$ is clear and appropriate, so please include a small sentence into the manuscript. At a suited place, please introduce something like: "The small signal at $B=0$ and $n_L=-4n_0$ can be attributed to a Seebeck contribution, originating from an asymmetry of the contacts."

And two small points:

In the abstract: "This was attributed to the emergence of orbital magnetic moment at the vHS." should be changed to:

"This was attributed to the emergence of orbital magnetic moment at the 2nd vHS at the hole side."

and:

"In contrast to this, an orbital magnetic moment will induce an energy shift under the application of the magnetic field; this is known as valley Zeeman splitting and has been demonstrated in gapped bilayer graphene"

here, change "magnetic field" to "perpendicular magnetic field".

Reviewer #3 (Remarks to the Author):

I appreciate the authors' explanations and expanding the manuscript based on reviewers' feedback.

In the last paragraph of the revised manuscript, I agree with the authors that orbital magnetism in twisted graphene structures are of great interest. The following works are more relevant to the discussion of orbital magnetism in those systems (also ABC-stacked trilayer graphene aligned with hBN), in my opinion.

Chen, et al. Tunable correlated Chern insulator and ferromagnetism in a moiré superlattice. *Nature*, 579(7797), 56–61.

Sharpe, et al. Emergent ferromagnetism near three-quarters filling in twisted bilayer graphene. *Science*, 365(6453), 605–608.

Serlin, et al. Intrinsic quantized anomalous Hall effect in a moiré heterostructure. *Science*, 367(6480), 900–903.

Polshyn, et al.. Nonvolatile switching of magnetic order by electric fields in an orbital Chern insulator. arxiv.2004.11353

Chen, et al. Electrically tunable correlated and topological states in twisted monolayer-bilayer graphene. arxiv.2004.11340

Thanks for reviewing our manuscript titled “Emergence of orbital angular moment at van Hove singularity in graphene/*h*-BN moiré superlattice”. Please find below, our replies to each of the comments.

Reviewer #1 (Remarks to the Author):

The manuscript has been revised according to the referees comments, and all questions of the referees have been answered convincingly. I recommend the publication of this manuscript in its current form.

Reply to reviewer #1) Thanks for reviewing our manuscript. We appreciate your time for reading our manuscript.

Reviewer #2 (Remarks to the Author):

The authors have revised a few key points in the second version of the manuscript and very thoroughly answered the question that I raised. The paper is publishable in NC and only minor comments need to be addressed.

Reply to reviewer #2) We are glad to hear that reviewer think our manuscript is publishable after only minor revision. Please find reply for your comments below.

Comment 1) It would be very beneficial for the reader, if a bit more of the answer to the reviewers were included in the manuscript in a concise form. Especially, this concerns the answer to my second comment, showing that the 2nd VHS at the hole side is also present in the paper by Kumar and the paper by Komatsu, even though it is not properly discussed there. To properly account for this, I suggest to adjust the following sentences:

"In this study, we observed that there was another point (i.e., a second vHS in the holeside located at the Y-point [red-dashed circle in Fig. 1(d)]) that exhibited a pronounced effect of inversion asymmetry; this point has not been experimentally investigated to date."

to:

"...of inversion asymmetry; this point has not been discussed to date."

and:

"The first vHSs on both the electron- and hole-side had previously been investigated [25]; here, we investigated the second vHS(h) which was not studied before."

to :

"...second vHS(h) which was observed [6,7] but not discussed before."

and:

"in Fig. 2(g). The dip-shape signal generated from the second vHS(h) exhibits significant splitting with increasing B, ..."

to:

"in Fig. 2(g). The dip-shape signal generated from the second vHS(h) exhibits significant splitting with increasing B (see also Refs. [6,7]), ..."

Reply for comment 1) Thanks for your suggestions. We modified the manuscript according to the suggestion given by the reviewer.

Comment 2) Concerning the answers to my comment on the temperature gradient (comment 5): The given answer is clear and detailed and I agree with the authors that it makes sense to publish this discussion in a separate paper. If this paper already exists, it would be nice to have the reference included in the revised manuscript.

Reply for comment 2) The paper that is discussing the temperature gradient is still in preparation; so we are not including it as a reference of the revised manuscript.

Comment 3) On the question on how important the etching of the hBN is: Please include the discussion and Figure R7 to the supporting information.

Reply for comment 3) We included the discussion on the etching of the h-BN as a Supplementary Note 3 in the revised manuscript.

Comment 4) Finally, the answer to comment 7 on the wiggle at B=0 is clear and appropriate, so please include a small sentence into the manuscript. At a suited place, please introduce something like: "The small signal at B=0 and $n_L = -4n_0$ can be attributed to a Seebeck contribution, originating from an asymmetry of the contacts."

Reply for comment 4) Thanks for suggesting this. We included new sentence "The small signal around $n_L = +2.20$ and $-2.25 \times 10^{12} \text{ cm}^{-2}$ at $B = 0 \text{ T}$ can be attributed to a Seebeck effect of SDP, originating from an asymmetry of the contacts." in the revised manuscript. Please note we slightly modified the sentence from the reviewer's suggestion. We believe this is more suitable to insert in the revised manuscript.

Comment 5) And two small points:

In the abstract: "This was attributed to the emergence of orbital magnetic moment at the vHS." should be changed to:

"This was attributed to the emergence of orbital magnetic moment at the 2nd vHS at the hole side."

and:

"In contrast to this, an orbital magnetic moment will induce an energy shift under the application of the magnetic field; this is known as valley Zeeman splitting and has been

demonstrated in gapped bilayer graphene"

here, change "magnetic field" to "perpendicular magnetic field".

Reply for comment 5)

We modified the manuscript according to the suggestion given by the reviewer. Since this modification makes word count of the abstract exceeds a length limit the journal, we adjusted few sentence in the abstract to meet the format of the journal. We are very glad if you could check the revised abstract of the manuscript.

Reviewer #3 (Remarks to the Author):

I appreciate the authors' explanations and expanding the manuscript based on reviewers' feedback.

In the last paragraph of the revised manuscript, I agree with the authors that orbital magnetism in twisted graphene structures are of great interest. The following works are more relevant to the discussion of orbital magnetism in those systems (also ABC-stacked trilayer graphene aligned with hBN), in my opinion.

Chen, et al. Tunable correlated Chern insulator and ferromagnetism in a moiré superlattice. *Nature*, 579(7797), 56–61.

Sharpe, et al. Emergent ferromagnetism near three-quarters filling in twisted bilayer graphene. *Science*, 365(6453), 605–608.

Serlin, et al. Intrinsic quantized anomalous Hall effect in a moiré heterostructure. *Science*, 367(6480), 900–903.

Polshyn, et al.. Nonvolatile switching of magnetic order by electric fields in an orbital Chern insulator. arxiv.2004.11353

Chen, et al. Electrically tunable correlated and topological states in twisted monolayer-bilayer graphene. arxiv.2004.11340

Reply to reviewer #3) Thanks for your suggestion. We included above-mentioned references in the revised manuscript.